# VideoChat-R1.5: Visual Test-Time Scaling to Reinforce Multimodal Reasoning by Iterative Perception

**Ziang Yan**[1,2,*]**, Xinhao Li**[3,2,*]**, Yinan He**[2,*]**, Zhengrong Yue**[2]**,**
Xiangyu Zeng[3,2], Yali Wang[4, 2], Yu Qiao[2], Limin Wang[3,2], Yi Wang[2,†]
[1]Zhejiang University [2]Shanghai AI Laboratory [3]Nanjing University
[4]Shenzhen Institutes of Advanced Technology, Chinese Academy of Sciences

`https://github.com/OpenGVLab/VideoChat-R1`

## Abstract

Inducing reasoning in multimodal large language models (MLLMs) is critical for achieving human-level perception and understanding. Existing methods mainly leverage LLM reasoning to analyze parsed visuals, often limited by static perception stages. This paper introduces Visual Test-Time Scaling (VTTS), a novel approach to enhance MLLMs' reasoning via iterative perception during inference. VTTS mimics humans' hierarchical attention by progressively refining focus on high-confidence spatio-temporal regions, guided by updated textual predictions. Specifically, VTTS employs an Iterative Perception (ITP) mechanism, incorporating reinforcement learning with spatio-temporal supervision to optimize reasoning. To support this paradigm, we also present VTTS-80K, a dataset tailored for iterative perception. These designs allows a MLLM to enhance its performance by increasing its perceptual compute. Extensive experiments validate VTTS's effectiveness and generalization across diverse tasks and benchmarks. Our newly introduced Videochat-R1.5 model has achieved remarkable improvements, with an average increase of over 5%, compared to robust baselines such as Qwen2.5VL-3B and -7B, across more than 15 benchmarks that encompass video conversation, video reasoning, and spatio-temporal perception.

## 1 Introduction

How to induce reasoning in multimodal large language models (MLLMs) becomes increasingly crucial in foundation models and multimodal understanding for general intelligence. Many believe [62, 47, 64, 50] it is one of the indispensable prerequisites to enable MLLMs with human-level perception and understanding, as people usually perceive surroundings with reflection, more than only parsing. Existing advances in MLLMs mostly concentrate on exploiting LLMs' reasoning to process parsed visuals deeply [55, 70, 67], or leveraging vision rules to elicit multimodal reasoning [47, 11]. Either of them explores rich causal dependencies in languages space to dig given evidences for more persuasive analysis and sound decisions. In this paper, inspired by humans' hierarchical attention with refinement [48, 43, 22], we study causal relations in vision for reasoning, delivering a learnable diagram to scale test-time compute of MLLMs by iterative perception.

To explicitly improve multimodal reasoning, existing approaches usually extend or enhance search space of LLMs and score them for the most reasonable one. Examples include Best-of-N (BoN) sampling [20], which selects the optimal output from multiple candidates via scoring; guided beam

---

*Equal contributions.
†Corresponding author.

39th Conference on Neural Information Processing Systems (NeurIPS 2025).

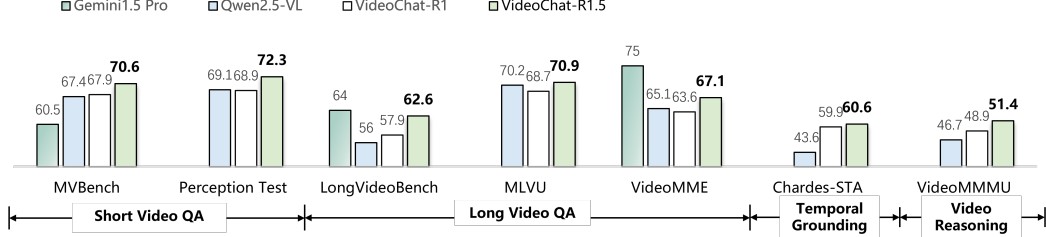

Figure 1: **Performance comparison with other models on multiple video benchmarks**

search [65], employing learned heuristics to refine decoding paths; and Monte Carlo Tree Search (MCTS) [14], using tree-based simulations for decision refinement. Meanwhile, some employ reinforcement learning (e.g. GRPO [21]) to improve LLM search space using visual rewards. In this regard, these methods only perceive once for subsequent analysis. Contrarily, V* [62] and Visual-CoT [47] show directly simulating humans' top-down perception also benefits reasoning by more accurate spatio-temporal understanding with fewer hallucinations. Note whether these aforementioned means perceive once or in several times, scaling their reasoning (or estimation parts) cannot remedy the perceived evidences in cases of bad luck.

In this work, we give a visual test-timing scaling (VTTS) approach for MLLMs, designed to enhance MLLMs' reasoning capabilities through iterative visual perception during inference. Inspired by the human strategy of progressively narrowing attention to relevant spatio-temporal regions, VTTS enables MLLMs to dynamically allocate computational focus to high-confidence visual regions across multiple iterations. This process is guided not only by the visual input but also by the evolving textual predictions, explicitly modeling dependencies between language and vision over time.

At the core of VTTS lies an Iterative Perception (ITP) mechanism that refines the model's understanding of multimodal inputs in stages. Each iteration involves predicting regions of interest (ROIs) based on prior reasoning, followed by reprocessing these regions to gather more detailed context. To support this multistage visual reasoning, we formulate a new objective that extends traditional autoregressive MLLM training to include visual dependency modeling. Recognizing current limitations in accurate modeling spatio-temporal locations only with autoregression, we further develop a reinforcement learning (RL) method based on Generalized Reward Policy Optimization (GRPO), enabling the model to learn spatio-temporal focus policies from existing spatio-temporal annotations.

To facilitate the training and evaluation of VTTS, we also introduce VTTS-80K, a dataset with fine-grained annotations for visual reasoning. VTTS-80K includes QA pairs enriched with annotated spatio-temporal cues and corresponding chains of thought, enabling models to learn how to identify and reason over critical visual segments.

Our experiments demonstrate that VTTS significantly improves MLLM performance from fundamental spatio-temporal perception, commonsense video question-answering, to complex multimodal reasoning by enabling targeted visual refinement at test time. On average, it enables Qwen2.5-VL-7B to increases in numeric results in aforementioned benchmarks by 5.4% and -3B by 6.3%. VTTS represents a step toward more human-like, adaptive perception in multimodal AI systems, bridging the gap between static visual processing and dynamic, context-driven attention.

- We propose a new test-time scaling method (VTTS) for MLLMs. We explicitly build visual dependencies with iterative perception (ITP) in MLLMs, simulating humans' progressive attention for ROIs. For ITP, we use reinforcement learning with a spatio-temporal verification to enhance the model's perception, leading to notable gains over strong baselines (e.g. TTS in LLMs).

- We propose the VTTS-80K dataset, which enables models to achieve iterative perception and reasoning capabilities through small-scale reinforcement fine-tuning.

- We verify the generalization of VTTS with different MLLMs in a spectrum of tasks over 15 benchmarks, including video QA, video reasoning, and spatio-temporal perception. We show consistent increases in these benchmarks for MLLMs with VTTS.

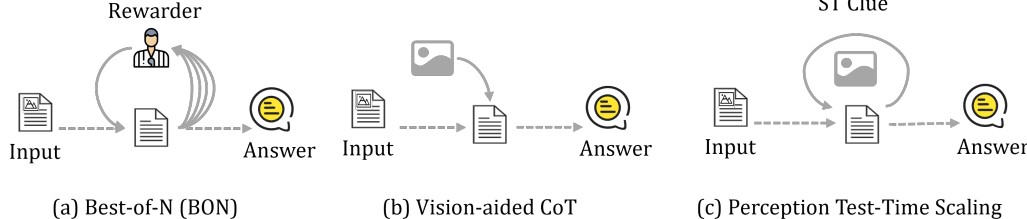

(a) Best-of-N (BON)  (b) Vision-aided CoT  (c) Perception Test-Time Scaling

Figure 2: **Schemes of Test time scaling methods.** BoN refers to generating N candidate items and selecting the best one, Vision-aided CoT involves incorporating visual information once into the reasoning process, and ITP entails iteratively generating spatiotemporal clues and selectively adding visual information to the reasoning based on these clues.

## 2 Related work

**Multimodal Large Language Model.** Large language models (LLMs) have been extended into the visual domain, leading to multimodal large language models (MLLMs). These models combine text and visual information for broader understanding capabilities. Early influential works like BLIP-2 [30], LLaVA [36], and mPLUG-Owl [71] successfully solved tasks such as image captioning and visual question answering. More recently, focus has been shifted towards extending MLLMs to handle video understanding, with methods such as VideoChat [31, 32], InternVideo [57–59], and Video-LLaMA [77, 10, 75]. These methods allow LLMs to interpret video by processing frame sequences and using video-focused instruction data. Additionally, some works [29, 4] have developed unified frameworks for both image and video understanding, aiming to better handle diverse visual inputs by connecting static and dynamic information.

Though MLLMs unified perceptual tasks by treating them as visual-based dialogues, making impressive strides in both open- [51, 4, 9] and closed- [49] models, they still lag behind traditional methods, which achieve superior performance through domain expertise and task-specific optimizations. Approaches such as TimeChat [46] and AllSeeing [53, 54] achieve high accuracy on a specific visual task. However, these strategies often compromise generalization capabilities, restricting their applicability to diverse scenarios. Frameworks like Videochat-TPO [68] achieved strong performance on both visual tasks and conversation by integrating task-specific modules. However, this approach introduces additional complexity and requires specialized components. Enhancing both task-specific performance and robust generalization capabilities remains a pivotal challenge in advancing the spatio-temporal reasoning capabilities of MLLMs.

**Test Time Scaling.** Scaling compute at inference has proven to be an effective strategy for enhancing reasoning without increasing model or training scale. Techniques such as Best-of-N [20], guided beam search [65], and Monte Carlo Tree Search [14] have achieved significant success in LLMs. Concerning MLLMs, test-time scaling has yet to be thoroughly explored. Previous work has primarily focused on emulating the approaches used in LLMs, such as extending the output length of MLLMs [47] or employing search-based methods during testing [60]. Note that these methods usually perform perception only once, barely leveraging the visual input dynamically during the extended reasoning process.

**Reinforcement Learning.** Reinforcement learning (RL) has proven to be a transformative approach for enhancing the reasoning capabilities of LLMs. Recent advancements, exemplified by OpenAI-o1 [25] and DeepSeek-R1 [21], have showcased remarkable progress in handling complex tasks, particularly in domains like mathematics and code generation, through the use of verifiable reward mechanisms. Building upon these advancements, researchers have extended RL techniques to MLLMs to enhance their visual reasoning performance. For instance, frameworks such as [80, 39, 78, 56, 79] leverage task-specific reward mechanisms to tackle challenges in fine-grained perception. However, the application of RL to video understanding remains relatively underexplored. Recent studies, such as VideoChat-R1 [34] and Video-R1 [13], have started to explore RL-based approaches for spatio-temporal reasoning, paving the way for advancements in this challenging domain.

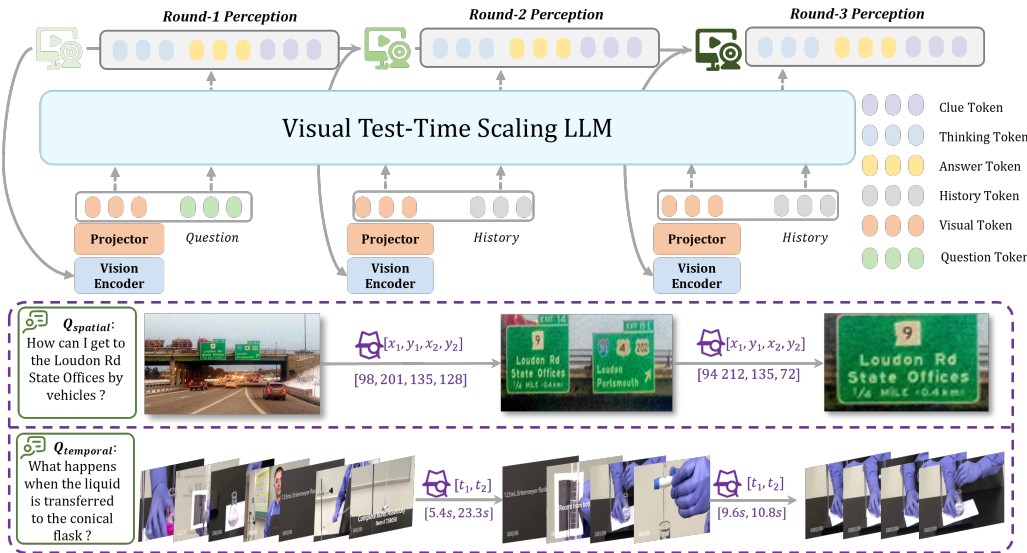

Figure 3: **Inference of iterative perception.**

## 3 Methodology

Visual test-time scaling (VTTS) aims to improve MLLMs' reasoning by introducing iterative vision computation in inference in a parametric manner. Inspired by human's coarse-to-fine processing for cues across visual scales, we make MLLMs simulate this progressive attention via predicting high-confidence spatio-temporal regions iteratively, explicitly building perceptual dependencies between processed visuals. The framework of VTTS is given in Figure 3.

Formally, the current learning of MLLMs mainly maximizes the conditional likelihood of the paired visual-question-answer pair $(\mathbf{V}, \mathbf{W}^q, \mathbf{W}^a)$ under the forward autoregressive factorization:

$$\mathcal{L} = -\log \sum_{i=1}^{T} P_\theta(\mathbf{W}_i^a | \mathbf{V}, \mathbf{W}_{<i}^a, \mathbf{W}^q), \tag{1}$$

where $\mathbf{W}_i^a$ and $T$ stand for the $i_{\text{th}}$ token and the ground truth length of the answer $\mathbf{W}^a$, respectively. $\mathbf{V}$ denotes video or image. For the better reasoning, existing solutions add predicted length or candidate number of the answer to cover and capture more complex word dependencies. Here, we extend the dependency modeling in reasoning from language to visuals as:

$$\mathcal{L}_v = -\log \sum_{k=1}^{K} \sum_{i=1}^{T} P_\theta(\mathbf{W}_{i,k}^a, \mathbf{V}_{k+1} | \mathbf{V}_k, \mathbf{W}_{<i,k}^a, \mathbf{W}^q), \quad \text{w.r.t.} \quad \mathbf{V}_{k+1} = \delta(\mathbf{V}_k | \mathbf{W}_k^a), \tag{2}$$

where $\delta(\mathbf{V}_k | \mathbf{W}^a)$ is a sampling function to generate a new visual $\mathbf{V}_{k+1}$ with focus from $\mathbf{V}_k$ based on the model's $k_{\text{th}}$ estimation $\mathbf{W}_k^a$.

To apply this likelihood computation concerning iterative visual handling in either MLLMs' training or testing is non-trivial, as it regards accurately narrowing down spatio-temporal scope based on textual predictions. Prior studies uncover the limitation of predicting spatio-temporal location, size, and shape using classical supervised text-based autoregressive formulation [74, 54], nevertheless typical MLLMs' training corpus (visual-question-answer pairs) mostly contain no specific numeric spatio-temporal descriptions. In this regard, we present a reinforcement learning based method to make this learning and inference tractable, as well as an accompanying dataset for its training.

### 3.1 Learning Iterative Perception with Reinforcement Fine-Tuning

We treat the visual sampling process $\delta(\mathbf{V}_k | \mathbf{W}^a)$ in Eqn. 2 as standard region of interest (ROI) selection. This demands MLLM's output to contain ROI's coordinates. To incorporate this explicit spatial prediction into MLLM's learning, we employ a reinforcement learning (RL) framework

based on Generalized Reward Policy Optimization [21] (GRPO). This approach enables the model to progressively refine its understanding of spatio-temporal contexts through repeated dynamic interactions with multimodal inputs. This learning is driven by both visual and textual supervisions as well as format requirements for their compatibility as:

$$\mathcal{R}(\theta) = \underbrace{\lambda_0 \cdot r_{\text{clue}}(\mathbf{W}_{\text{clue}}^a, \hat{\mathbf{W}}_{\text{clue}}^a)}_{\text{Spatio-Temporal Awareness}} + \underbrace{\lambda_1 \cdot r_{\text{ans}}(\mathbf{W}^a, \hat{\mathbf{W}}^a)}_{\text{Answer Supervision}} + \underbrace{\lambda_2 \cdot r_{\text{fmt}}(\mathbf{W}^a, \hat{\mathbf{W}}^a)}_{\text{Output Format Supervision}}, \tag{3}$$

where $\lambda_0$, $\lambda_1$, and $\lambda_2$ are balancing coefficients, and $r_{\text{clue}}$, $r_{\text{ans}}$, and $r_{\text{fmt}}$ are rewards for RL. $\mathbf{W}_{\text{clue}}^a$ and $\hat{\mathbf{W}}_{\text{clue}}^a$ stand for the ground truth and the predicted spatio-temporal coordinates tuples, respectively. Specifically, $r_{\text{clue}}$ verifies the spatio-temporal alignment between two spatio-temporal segments, and we exploit intersection over union (IoU) for its implementation. $r_{\text{ans}}$ and $r_{\text{fmt}}$ check the answer accuracy and format, respectively. If the generated answer or its format matches the ground truth then the reward scores 1, otherwise 0. For $r_{\text{fmt}}$, it uses regular expressions to verify adherence.

We also make tries optimize Eqn. 2 with standard supervised fine-tuning, letting MLLMs generate textual answers as well as numerical regional descriptions for focus. Experiments in Sec 4 show this hardly benefits finding regions of interest and improving subsequent reasoning.

**Inference with Test-Time Scaling.** Our inference leverages an Iterative Perception (ITP) strategy, which enables the model to progressively refine its understanding of spatio-temporal contexts through multiple perception cycles. This approach is particularly effective for handling complex multimodal inputs, where salient information is often embedded within specific regions or time segments. By iteratively focusing on these key areas while maintaining a global context, the model achieves a more nuanced and accurate interpretation of the input data.

The first step in the ITP strategy involves standard processing akin to that used in conventional MLLMs. For video inputs, this entails uniform frame sampling across the entire sequence to capture a broad overview of the temporal dynamics. For image inputs, the model processes the full image to establish an initial understanding of the spatial layout. During this stage, the model generates preliminary reasoning steps, identifies relevant spatio-temporal clues (e.g., specific time intervals in videos or bounding boxes in images), and outputs an initial answer. While this initial pass provides a baseline understanding, it may lack sufficient granularity to address complex queries accurately.

Subsequent iterations implement a differential processing strategy, which dynamically reallocates computational resources to focus on the identified spatio-temporal cues while preserving the broader context. For videos, this involves dense frame sampling within the identified temporal segments where critical events or actions are likely to occur while applying sparse sampling elsewhere to reduce redundancy and computational overhead. For images, the model concurrently processes both the full image and cropped versions of the identified spatial regions. This dual-input approach ensures that the model retains a holistic view of the scene while zooming in on specific areas of interest. Simultaneously, the model leverages the think process from the preceding step as a linguistic prior to inform and refine its subsequent reasoning.

This hierarchical approach not only improves the model's ability to extract meaningful features but also enhances its capacity to reason about complex interactions between objects or events.

## 3.2 VTTS 80k Data Generation

To enable the learning of Iterative Perception with RFT, we build VTTS-80K dataset from existing ones. This dataset aims to enhance spatio-temporal awareness, enabling models to locate relevant visual clues and perform reasoning based on these observations, thereby facilitating iterative perceptual refinement. Our annotations provide rich supervision through three parts: QA pairs, relevant spatio-temporal segments (critical regions or time intervals for the question), and the thought process to reach the answer. As detailed in Fig 4, VTTS-80K comprises three sections targeting different skills: VTTS-QA(VideoQA and Image Reasoning) and for reasoning ability, VTTS-TEMP for temporal understanding, and VTTS-SPATIAL for spatial awareness. Additional dataset statistics are provided in the appendix.

The construction of VTTS-80K follows a systematic pipeline aimed at producing high-quality reasoning annotations. The initial step involves LLM-based verification: DeepSeek assesses whether the QA pairs are consistent with the corresponding image/video caption. Simultaneously, it evaluates

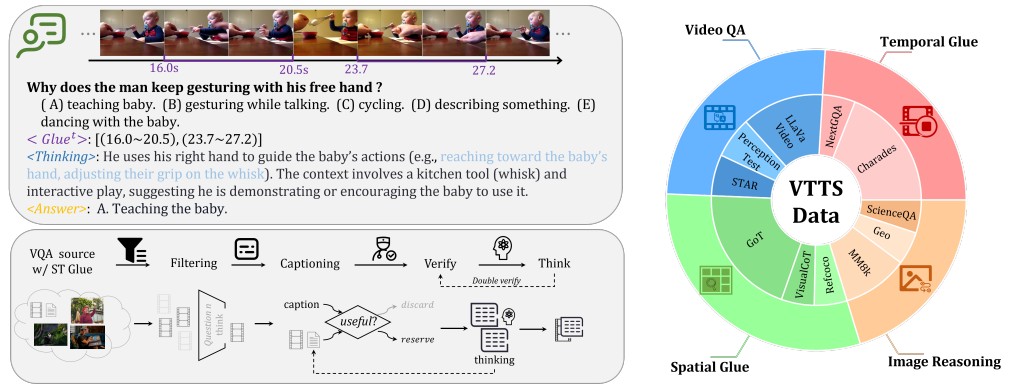

(a) *Pipeline of Data Generation*          (b) *Data Distribution of VTTS RL Training*

Figure 4: **VTTS-80K dataset generation pipeline and data distribution.**

the relevance of potential spatio-temporal cues by comparing their descriptions (e.g., captions of specific segments) against the overall image/video context and the QA pair. This dual verification filters out inconsistent QA pairs and irrelevant cues, resulting in a refined set called Filtered QA (containing validated QA pairs and associated relevant cues). Next, the Filtered QA is fed into DeepSeek for Reasoning, which generates step-by-step CoT explanations. This step breaks down the reasoning needed to arrive at the answer, explicitly linking steps to relevant multimodal information (like object interactions or temporal changes) within the image/video. Finally, a VLM ranks these generated QA-CoT pairs based on their clarity, relevance, and consistency, producing the final VTTS-80K dataset.

## 4   Experiments

**Implementation.**   We apply VTTS to the latest Qwen series like Qwen2.5-VL-7B and Qwen2.5-VL-3B using VTTS-80K dataset with reinforcement fine-tuning(RFT). Training is performed with a learning rate of 2e-6 and a batch size of 16. The reward function comprise three components: format reward, clue reward (quantified by the IoU of visual clues), and answer reward. At inference time, the default number of iterative perception (ITP) iterations is set to 3.

**Benchmarks.**   To comprehensively evaluate the general capabilities of our models, we conduct experiments across a diverse suite of benchmarks. For video perception, we report results on MVBench [32] and PerceptionTest [44], which assesses fine-grained temporal understanding, including action types, sequences, and movement directions. General video understanding is assessed using VideoMME [16], consisting of videos of short, medium, and long durations. In the long-video domain, we evaluate performance on MLVU [81], LVBench [52], and LongVideoBench [61]. Video-based knowledge modeling is quantified using the VideoMMMU [23] benchmark. To evaluate the model's visual-spatial intelligence, we adopt the VSIBench [69]. Furthermore, we specifically assess spatio-temporal grounding capabilities using a range of detection and temporal grounding datasets.

### 4.1   General Understanding Evaluation

**Multimodal Video Understanding.**   To comprehensively evaluate the video understanding capabilities of our model, we conduct assessments on multiple video benchmarks, comparing performance under both single-perception and multi-perception settings. The 7B and 3B models of VideoChat-R1.5 achieve accuracies of 70.6% and 68.1%, respectively, on MVbench [32], outperforming the baseline by 2.2% and 1.6%. As shown in Tab 1, on VideoMME[16], our 7B and 3B models surpass Qwen2.5-VL by 2% and 2.7%, respectively, with iterative perception outperforming single-step perception by 1.9% and 1.2%. In benchmarks focused on long video understanding, such as LongVideoBench [61], LVbench[52], and MLVU [81], our 7B model achieves gains of 6.6%, 0.7%, and 1.1%, respectively, without using additional long-video training data. On the video knowledge modeling benchmark VideoMMMU [23], our 7B and 3B models outperform the baseline by 2.2% and 1.5%, respectively.

Table 1: **Performance on video question-answering benchmarks.** The suffix "-s" denotes single-step perception, while "-m" indicates three iterations of iterative perception.

| Model | Size | MVBench [32] | PerceptionTest [44] | LongVideoBench [61] | VideoMMMU [23] | MLVU [81] | VideoMME [15] | LVBench [52] |
|---|---|---|---|---|---|---|---|---|
| Average duration (sec) | | 16 | 23 | 473 | 507 | 651 | 1010 | 4101 |
| *Proprietary Models* | | | | | | | | |
| GPT4-V [3] | - | 43.7 | - | 59.1 | - | 49.2 | 59.9 | - |
| GPT4-o [2] | - | 64.6 | 72.2 | 66.7 | 61.2 | 64.6 | 71.9 | 30.8 |
| Gemini-1.5-Pro [49] | - | 60.5 | 71.2 | 64.0 | 53.9 | - | 75.0 | 33.1 |
| *Open-Source MLLMs* | | | | | | | | |
| InternVL2.5 [8] | 2B | 68.8 | - | 46.0 | - | 61.4 | 51.9 | - |
| VideoChat-Flash[33] | 2B | 70.0 | 70.5 | 58.3 | - | 65.7 | 57.0 | 42.9 |
| QWen2.5-VL-3B | 3B | 67.0 | 66.9 | 54.2 | 42.3 | 68.2 | 61.5 | 43.3 |
| **VideoChat-R1.5-3B-S** | 3B | 68.2 | 67.6 | 58.4 | 42.7 | 68.4 | 63.0 | 45.2 |
| **VideoChat-R1.5-3B-M** | 3B | 68.6(+1.6) | 68.3(+1.4) | 59.6(+5.4) | 43.8(+1.5) | 69.7(+1.5) | 64.2(+2.7) | 46.6(+3.3) |
| LLava-OneVision [29] | 7B | 56.7 | 57.1 | 56.3 | 34.4 | 64.7 | 58.2 | - |
| InternVL2.5 [8] | 7B | 72.0 | 51.5 | 60.0 | - | 68.9 | 64.2 | 38.4 |
| QWen2.5-VL-7B | 7B | 68.4 | 70.5 | 56.0 | 46.7 | 70.2 | 65.1 | 45.3 |
| **VideoChat-R1.5-7B-S** | 7B | 70.6 | 71.3 | 61.4 | 49.6 | 70.1 | 65.2 | 46.3 |
| **VideoChat-R1.5-7B-M** | 7B | 70.6(+2.2) | 72.3(+1.8) | 62.6(+6.6) | 51.4(+4.7) | 70.9(+0.7) | 67.1(+2.0) | 48.4(+3.1) |

Moreover, iterative perception consistently surpasses single-step perception across these evaluations. These validate the effectiveness of ITP and show the initial scaling effects.

**Grounded VideoQA.** Grounded video QA task requires the model to not only provide accurate answers regarding videos, but also identify the specific temporal segments that support those answers. This task highlights the need for joint reasoning between semantic understanding and temporal context, leading to accurate and interpretable predictions. We evaluate our model on two grounded video QA benchmarks: NextGQA [63] and ReXTime [7]. In NextGQA [63], Acc@IoP@0.5 denotes the proportion of questions where IoP is bigger than 0.5 and the model correctly answers the multiple-choice question, while Acc@GQA reflects the accuracy on questions where both IoP is bigger than 0.5 and the QA is correct. VideoChat-R1.5 demonstrates non-trivial increase over the baseline in both QA and IoP metrics across both 3B and 7B parameter scales. These gains highlight the effectiveness of our reinforcement learning in enhancing fine-grained temporal reasoning and grounding capabilities. Furthermore, on ReXTime [7],VideoChat-R1.5-7B surpasses GPT-4o [2] across all metrics, showcasing the model's strong QA capabilities and grounding performance. The superior performance on ReXTime underscores the model's ability to handle complex and diverse video content while maintaining high accuracy in both QA and grounding tasks.

Table 2: **Performance on NextGQA [63].**

| Model | Acc@IoP@0.5 | Acc@GQA | mIoP | IoP@0.3 | IoP@0.5 |
|---|---|---|---|---|---|
| VIOLETv2 [17] | 54.9 | 12.8 | 23.6 | 25.1 | 23.3 |
| SeViLA [73] | 72.5 | 16.6 | 29.5 | 34.7 | 22.9 |
| LangRepo [26] | 59.6 | 17.1 | 31.3 | - | 28.7 |
| VideoStreaming [45] | 57.4 | 17.8 | 32.2 | - | 31.0 |
| LLoVi [76] | 65.9 | 24.3 | 37.3 | - | - |
| VideoChat-TPO [68] | 77.7 | 25.5 | 35.6 | 47.5 | 32.8 |
| QWen2.5-VL-3B | 70.3 | 15.5 | 24.9 | 33.0 | 22.1 |
| **VideoChat-R1.5-3B** | 76.5 | 48.9 | 62.3 | 71.2 | 63.9 |
| QWen2.5-VL-7B | 72.7 | 42.3 | 54.0 | 62.6 | 54.5 |
| **VideoChat-R1.5-7B** | 79.9 | 61.9 | 74.9 | 82.7 | 77.6 |

Table 3: **Performance on RexTime [7].**

| Model | Acc | Acc@IoU@0.5 | mIoU | IoU@0.3 | IoU@0.5 |
|---|---|---|---|---|---|
| UniVTG [35] | - | - | 28.2 | 41.4 | 26.9 |
| CG-DETR [42] | - | - | 23.9 | 31.3 | 16.7 |
| VideoChat-TPO [68] | - | - | - | - | - |
| Claude3-Opus [1] | 68.7 | 13.7 | 28.4 | 35.7 | 25.0 |
| GPT4o [2] | 73.7 | 28.7 | 36.3 | 45.3 | 34.0 |
| QWen2.5-VL-3B | 53.6 | 1.6 | 7.4 | 7.5 | 3.1 |
| **VideoChat-R1.5-3B** | 62.1 | 12.1 | 21.3 | 30.8 | 16.6 |
| QWen2.5-VL-7B | 70.4 | 20.9 | 29.6 | 38.8 | 25.2 |
| **VideoChat-R1.5-7B** | 74.8 | 38.1 | 45.8 | 61.8 | 46.4 |

## 4.2 Spatial-Temporal Tasks

Table 4: **Fine-tuning results of temporal grounding.**

| Method | Charades-STA [19] | | | |
|---|---|---|---|---|
| | mIoU | R@0.3 | R@0.5 | R@0.7 |
| InternVideo2 [58] | - | - | 70.0 | **48.9** |
| TimeSuite [74] | - | 79.4 | 67.1 | 43.0 |
| QWen2.5-VL-3B | 38.8 | 65.3 | 39.2 | 20.8 |
| **VideoChat-R1.5-3B** | 50.8 | 74.9 | 58.6 | 30.9 |
| QWen2.5-VL-7B | 43.6 | 76.1 | 42.9 | 26.2 |
| **VideoChat-R1.5-7B** | 60.6 | 82.8 | 71.6 | 48.3 |

Table 5: **Zero-shot results of temporal grounding.**

| Method | QVHighLight [28] | | | | ActivityNet [6] | | | |
|---|---|---|---|---|---|---|---|---|
| | mIoU | R@0.3 | R@0.5 | R@0.7 | mIoU | R@0.3 | R@0.5 | R@0.7 |
| Videochat-TPO [68] | 40.7 | 56.9 | 40.1 | 22.0 | 27.6 | 42.6 | 26.3 | 13.0 |
| TimeSuite [74] | 44.8 | 57.2 | 45.1 | 27.0 | - | - | - | - |
| **QWen2.5-VL-3B** | 21.3 | 34.1 | 23.0 | 13.9 | 10.9 | 17.9 | 10.6 | 4.2 |
| **VideoChat-R1.5-3B** | 33.9 | 46.8 | 34.4 | 21.9 | 23.9 | 35.0 | 18.9 | 8.7 |
| **QWen2.5-VL-7B** | 30.6 | 42.9 | 32.0 | 20.1 | 19.1 | 25.5 | 13.4 | 6.1 |
| **VideoChat-R1.5-7B** | 52.7 | 71.4 | 55.8 | 38.4 | 35.5 | 52.4 | 32.3 | 16.8 |

**Temporal Grounding.** Temporal grounding refers to the task of localizing target temporal segments in a video that correspond to a given natural language query. This task demands precise alignment between linguistic semantics and some clip of a video. As shown in Table 4 and 5, VideoChat-R1.5 model demonstrates compelling temporal grounding performance in both zero-shot and fine-tuned settings. In fine-tuned settings, both VideoChat-R1.5-3B and VideoChat-R1.5-7B model

variants significantly outperform all baselines, achieving SOTA performance among existing MLLMs. Notably, on the R@0.5 metric, our 7B model achieves a score of 71.6, surpassing specialized expert models based on InternVideo2-6B [58], which are explicitly designed for temporal grounding tasks. Furthermore, in the zero-shot setting, our model exhibits substantial improvements over previous baselines, demonstrating remarkable generalization capability without task-specific training. This zero-shot performance suggests the potential for deploying our model in real-world applications with limited labeled data. This zero-shot performance suggests that our model possesses strong temporal awareness, enabling it to accurately localize relevant time segments through iterative perception.

Table 6: **Results of spatial grounding.**

| Method | RefCOCO [72] | | | RefCOCO+ [72] | | | RefCOCOg [41] | |
|---|---|---|---|---|---|---|---|---|
| | val | tesetA | testB | val | tesetA | testB | val | test |
| MDETR [27] | 86.8 | 89.6 | 81.4 | 79.5 | 84.1 | 70.6 | 81.6 | 80.9 |
| Videochat-TPO [68] | 85.9 | 90.8 | 81.3 | 80.2 | 85.1 | 71.7 | 79.4 | 81.3 |
| Grounding-DINO-L [37] | 90.6 | 93.2 | 88.3 | 82.7 | 88.9 | 75.9 | 96.2 | 87.0 |
| Qwen2.5-VL-3B | 89.1 | 91.7 | 84.0 | 82.4 | 88.0 | 74.1 | 85.2 | 85.7 |
| **VideoChat-R1.5-3B** | 90.3 | 92.8 | 85.3 | 83.4 | 88.9 | 75.8 | 86.2 | 86.3 |
| Qwen2.5-VL-7B | 90.0 | 92.5 | 85.4 | 84.2 | 89.1 | 64.9 | 87.2 | 87.2 |
| **VideoChat-R1.5-7B** | 91.1 | 93.6 | 86.7 | 84.7 | 90.3 | 77.8 | 87.6 | 87.7 |

Table 7: **Results of object tracking.** We use 8-frame inputs for both training and evaluation.

| Method | GoT [24] | | |
|---|---|---|---|
| | AO | SR@0.5 | SR@0.75 |
| SiamFC [5] | 34.8 | 35.3 | 6.8 |
| QWen2.5-VL-3B | 10.7 | 1.5 | 0 |
| **VideoChat-R1.5-3B** | 45.9 | 40.3 | 7.6 |
| QWen2.5-VL-7B | 12.6 | 1.1 | 0 |
| **VideoChat-R1.5-7B** | 52.2 | 45.7 | 9.8 |

**Spatial Grounding.** To evaluate the model's fine-grained localization capability, we experiment on the spatial grounding task, where models follow textual descriptions to give the corresponding bounding boxes on RefCOCO[72], RefCOCO+[72], RefCOCOg[41]. As shown in Table 6, we compare our VideoChat-R1.5 model with both MLLMs and specialized expert models such as G-DINO [37]. Remarkably, our VideoChat-R1.5 model achieves performance improvements over the strong QwenVL baseline, despite being trained on only 16k in-domain samples. Moreover, it outperforms several expert models that are fine-tuned on much larger datasets. This highlights the model's strong spatial awareness, providing a solid foundation for iterative perception and complex visual reasoning tasks.

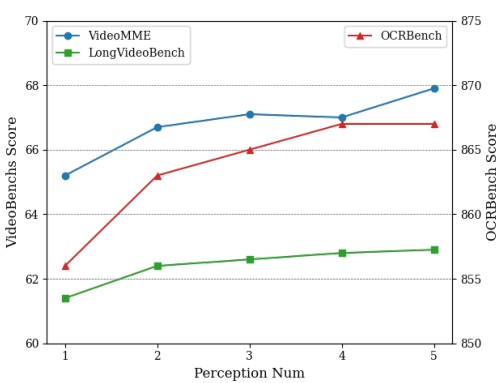

Figure 5: **Ablation on perception times.**

**Tracking.** In tracking, the model receives the object coordinates from the first frame of a video and is expected to output the corresponding coordinates for the remaining frames. Due to the input length limitations of MLLMs, we uniformly sample 8 frames from each video and generate predictions for all of them. We evaluate our model on the widely used tracking benchmark GOT-10k [24]. As shown in Table 7, our model demonstrates significant improvements over the baseline in terms of its ability to track objects through continuous motion. This advancement reflects a notable leap in the model's spatio-temporal localization capabilities, which is highly beneficial for fine-grained perception in video understanding.

### 4.3 Ablation

In this section, we analyze the effectiveness of iterative perception, validate the contributions of different components in our VTTS-80K dataset, and demonstrate the superiority of Progressive Reinforcement Learning over SFT for enhancing spatio-temporal understanding and reasoning.

**Scaling Law with Perception.** To validate the effectiveness of the iterative perception approach, we conducted ablation experiments on long video understanding and image perception datasets, focusing on how model performance scales with the number of perception iterations. As shown in Figure5, the model's performance exhibits a clear scaling trend as the number of perception steps increases. Specifically, on VideoMME [15], the score improves from 65.2 to 67.9, demonstrating a steady gain in multimodal reasoning capabilities. Similarly, on LongVideoBench[61], the performance rises

from 61.4 to 62.9, highlighting the model's enhanced ability to handle complex, temporally extended content. Additionally, on OCRBench [38], the score increases from 856 to 866, reflecting improved fine-grained visual perception. These consistent improvements across diverse tasks underscore a perception scaling law , where iterative refinement of spatio-temporal understanding leads to measurable performance gains. The results suggest that, with each additional perception iteration, the model becomes increasingly adept at capturing subtle visual cues, leveraging insights from prior steps, and integrating these into a more comprehensive and accurate representation of the input.

Table 8: **Ablation studies on training strategy and data.**

| Model | Training strategy | Training data | | | | Charades-STA | RefCOCO | | | Nextgqa | | OCRBench | VSIBench | VideoMME |
|---|---|---|---|---|---|---|---|---|---|---|---|---|---|---|
| | | S-Grounding | T-Grounding | I-MCQ | V-MCQ | mIoU | val | testA | testB | Acc@GQA | mIoP | Score | Avg | Avg |
| *Training Data Ablation* | | | | | | | | | | | | | | |
| Qwen2.5-VL-7B | - | | | | | 43.6 | 90.0 | 92.5 | 85.4 | 70.4 | 20.9 | 856 | 39.2 | 65.1 |
| Qwen2.5-VL-7B | RFT | | ✓ | ✓ | ✓ | 59.1 | 89.7 | 92.6 | 85.1 | 79.1 | 61.0 | 851 | 39.7 | 65.9 |
| Qwen2.5-VL-7B | RFT | ✓ | | ✓ | ✓ | 43.1 | 91.0 | 94.0 | 86.5 | 72.4 | 21.3 | 862 | 39.8 | 65.2 |
| Qwen2.5-VL-7B | RFT | ✓ | ✓ | | | 60.0 | 91.3 | 93.8 | 86.4 | 75.6 | 58.6 | 865 | 40.3 | 64.8 |
| Qwen2.5-VL-7B | RFT | | | ✓ | ✓ | 42.6 | 89.9 | 92.4 | 85.9 | 76.1 | 21.9 | 857 | 39.1 | 65.4 |
| *Training Strategy Ablation* | | | | | | | | | | | | | | |
| Qwen2.5-VL-7B | SFT | ✓ | ✓ | ✓ | ✓ | 54.9 | 88.3 | 90.2 | 83.6 | 70.8 | 31.6 | 815 | 37.8 | 63.9 |
| Qwen2.5-VL-7B | RFT | ✓ | ✓ | ✓ | ✓ | **60.6** | **91.1** | **93.6** | **86.7** | **79.9** | **61.9** | **865** | **40.6** | **67.1** |

**Temporal-Spatial Data.** We conducted ablation experiments on the training data to validate the effectiveness of each component of our VTTS-80K dataset. As shown in Table 8, it is observed that the Spatial Grounding data and Temporal Grounding data significantly enhance performance on spatial fine-grained tasks and temporal fine-grained tasks, respectively. Additionally, the chat data prove beneficial for long video understanding, image perception, and spatial reasoning tasks.

**SFT vs. RL.** To rigorously validate our training paradigm, we contrast Progressive Reinforcement Learning with canonical SFT. As summarized in Table 8, reinforcement learning yields double-digit gains on temporally- and spatially-grounded localization benchmarks, whereas SFT not only underperforms but actually erodes the backbone model's original competence. This degradation underscores a well-known pathology of SFT: when the optimization objective is reduced to next-token likelihood, the model overfits to the annotation distribution and forgets useful pre-training priors. Reinforcement learning, by contrast, delivers uniform improvements on both in-domain and zero-shot out-of-domain splits, demonstrating that iterative reward shaping preserves—and systematically enhances—multi-granular perception capabilities. The core limitation of SFT is its reliance on cross-entropy over discretized bins. For inherently continuous targets—timestamps, durations, object counts—this produces a supervision signal that is oblivious to metric structure: any deviation from the exact ground-truth bin is penalized equally, regardless of numerical proximity. GRPO replaces this categorical loss with an L1 penalty directly defined on the continuous quantity space. The resulting reward landscape is fine-grained and metric-aware: infinitesimal deviations incur infinitesimal penalties, while larger errors are penalized proportionally. Consequently, the policy learns to regress implicit numerical clues with sub-second (or sub-pixel) precision, a prerequisite for downstream tasks that depend on accurate temporal or quantitative reasoning.

## 5   Conclusions

In this paper, we present VTTS, a novel test-time scaling framework for MLLMs that enhances reasoning through iterative visual perception. Inspired by human-like progressive attention, VTTS dynamically refines focus on key spatio-temporal regions, improving interpretability and reasoning accuracy. By introducing the VTTS-80K dataset, we enable MLLMs to learn iterative perceptual refinement, supported by reinforcement learning techniques tailored for multimodal contexts.

Extensive experiments across more than 15 benchmarks demonstrate that VTTS notably outperforms strong baselines in video conversation, image reasoning, and spatio-temporal perception tasks, highlighting the effectiveness of iterative perception in enhancing multimodal reasoning.

**Limitations.** Currently, VTTS has been validated solely in the domain of visual processing, and its inference process is not fully optimized to leverage visual-language priors or existing caching techniques during iterative computation. Extending VTTS to additional modalities and further optimizing its engineering implementation are left as directions for future work.

## Acknowledgements

This work is supported by the National Key R&D Program of China (Grant No. 2022ZD0160102), and the Jiangsu Frontier Technology Research and Development Program (Grant No. BF2024076).

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

# Appendix

In the Appendix, we provide additional details to support the main content of the paper. First, we expand on the specifics of the VTTS-80K dataset, including its composition and annotation types. Next, we detail the training and inference parameters used for the model. In the Case Study section, we present multiple cases of model reasoning, showcasing both successful examples and an analysis of failure cases to provide a comprehensive understanding of the model's performance. Finally, we include a discussion of the paper's limitations, broader societal impacts, and safeguards to address potential risks, ensuring a responsible and ethical approach to the development and deployment of the proposed method.

# 6 Trade-off Between Performance and Test Time.

Table 9: **Trade-off between VTTS performance and inference time**

| Model | Inference Type | Image (MMVet) | | Video (VideoMME) | |
|---|---|---|---|---|---|
| | | Infer Time | Performance | Infer Time | Performance |
| LLaVA-CoT [66] | CoT | 1.415 | 60.3 | - | - |
| VideoChat-R1 [34] | CoT | - | - | 13.535 | 62.4 |
| Qwen2.5VL-7B (Baseline) | Direct Output | 1.085 | 64.9 | 11.265 | 64.4 |
| Qwen2.5VL-7B (Baseline) | CoT | 1.465 | 63.2 | 14.905 | 61.3 |
| VideoChat-R1.5-7B | Single Infer | 1.355 | 67.2 | 13.215 | 65.2 |
| VideoChat-R1.5-7B | Multi Infer | 2.975 | **68.3** | 30.095 | **67.1** |

As illustrated in Table 9, the proposed VideoChat-R1.5 model achieves a favorable trade-off between computational efficiency and performance when compared to both the direct output baseline and other CoT reasoning methods. Specifically, VideoChat-R1.5-7B (Single Infer) attains an average inference time of 1.35 seconds on the MMVet benchmark, which is only marginally higher than the 1.08 seconds of the direct output baseline. This inference latency is comparable to other CoT-based approaches, such as LLaVA-CoT (1.41s) and MM-Eureka (1.39s), while significantly surpassing them in terms of performance (67.2%). Furthermore, although the multi-inference strategy introduces additional computational overhead, it yields a performance gain of 1.1%. On the VideoMME dataset, VideoChat-R1.5-7B (Single Infer) requires 13.21 seconds for inference, slightly exceeding the baseline's 11.26 seconds, yet achieves a notably higher accuracy of 65.2%, outperforming most existing long-chain reasoning methods. These results demonstrate that VideoChat-R1.5 not only maintains competitive inference efficiency but also delivers superior performance, particularly in complex multimodal reasoning tasks.

# 7 Details of VTTS-80K

## 7.1 Dataset Annotation

As shown in Tab. 10, the VTTS-80K dataset is composed of four key components: VideoQA, Temporal Clue, Image Reasoning, and Spatial Clue, which are carefully curated from existing datasets to support iterative perception. The dataset includes five types of annotations: question, option, think, clue, and answer. Among these, the clue annotation plays a critical role, as it encapsulates the essential information required to answer the question. This annotation is further divided into two subcategories: temporal clue, which captures time-related information, and spatial clue, which focuses on location or spatial relationships within the input data.

Notably, not all datasets contain every type of annotation. To address this, our training process dynamically selects the corresponding reward function based on the annotations available in each dataset. This ensures that the model can effectively leverage the diverse information provided by different subsets of the dataset while maintaining robustness across tasks with varying levels of annotation completeness. By aligning the reward mechanism with the available annotations, our approach maximizes the utility of the data and enables efficient learning even when certain types of annotations are missing.

Table 10: VTTS-80K data distribution and examples.

| Data Type | Data Source | Data Example |
|---|---|---|
| Video QA | LLavaVideo [29] | **Question:** Why does the text 'FROM ACCLAIMED DIRECTORS WERNER HERZOG AND CLIVE OPPENHEIMER' appear?
**Options:** A. To introduce the festive atmosphere, B. To show the location of the observatory domes, C. To highlight the geological patterns of the crater, D. To indicate the directors of the film
**Think:** The text "FROM ACCLAIMED DIRECTORS WERNER HERZOG AND CLIVE OPPENHEIMER" appears in the video to indicate the directors of the film. This is a standard practice in cinematic trailers to highlight the creative team behind the production, especially when the directors are renowned figures like Werner Herzog (a celebrated filmmaker) and Clive Oppenheimer (a producer known for documentaries). Their names serve to attract audiences familiar with their previous works and establish credibility for the film's quality.
**Answer:** D. To indicate the directors of the film |
| | PerceptionTest [44] | **Question:** What action not related to making tea did the person do?
**Options:** A. clapping hands B. stacking pots on the stove C. putting a sugar cube into the teabox.
**Think:** In the video, the person prepares tea by opening a tea box, placing a tea bag in a mug, pouring hot water, adding sugar, and stirring. There is no interaction with the stove or pots; the focus is solely on tea preparation.
**Answer:** C. The cup/glass/bottle. |
| | Star [62] | **Question:** What happened after the person tidied up the clothes?
**Options:** A. Took the towel. B. Opened the refrigerator. C. Put down the shoe. D. Closed the laptop.
**Think:** After tidying up the clothes, the person picked up the towel from the floor and carried it with them as they exited the room. This action directly follows the tidying process and aligns with the sequence shown in the video.
**Answer:** A. Took the towel. |
| Temporal Clue | Charades [19] | **Question:** To accurately pinpoint the event "the person takes a box out" in the video, determine the precise time period of the event.
**Think:** The event "the person takes a box out" occurs after entering the room and bending down to retrieve the box from a low shelf or the floor, then exiting the room while carrying the box. The sequence is: Enters the room. Bends down to pick up the box. Stands up with the box. Exits the room.
**Clue:** [6.0, 11.9] |
| | NextGQA [63] | **Question:** Why does the woman in blue swipe her eyes near the end of the video?
**Options:** A. remove makeup, B. wipe tears, C. dance together, D. make face, E. put cream on man s face
**Think:** The woman in blue is seen covering her face with her hands and then wiping her eyes. This action suggests she might be emotional or crying. The options provided do not include 'remove makeup' or 'put cream on man's face', so we can eliminate those. 'Dance together' and 'make face' do not fit the context of the scene. Therefore, the most likely reason is that she is wiping tears.
**Clue:** [11.6, 19]
**Answer:** B. wipe tears |
| Image Reasoning | Multimodal-Open-R1-8k [12] | **Question:** Based on the image, calculate the ratio of the base of the largest triangle to the base of the smallest triangle.
**Options:** A. 3 to 2. B. 5 to 3. C. 2 to 1. D. 9 to 5
**Think:** To determine the ratio of the base of the largest triangle to the smallest triangle, we analyze the similar triangles formed in the image. The largest triangle is $\triangle$ACE with base AE=30, and the smallest triangle is $\triangle$CBD with base BD. The horizontal lines divide the sides proportionally: on the left side CA, the segments are 3 and 3 (total 6), and on the right side CE, the segments are 5 and 5 (total 10). These divisions simplify to a ratio of 1:1, indicating similar triangles with a scale factor of 2:1. Since the triangles are similar, the base of the smallest triangle is half the base of the largest triangle: BD=30× 1/2=15. Therefore, the ratio of the largest base to the smallest base is 30:15=2:1.
**Answer:** C. 2 to 1. |
| | Geo [18] | **Question:** What is the degree measure of angle PEF in the quadrilateral ABCD figure, where AD is equal to BC, and point P serves as the midpoint of diagonal BD, while E and F represent the midpoints of AB and CD, if angle EPF equals 130 degrees?
**Options:** A. 25° B. 30° C. 35° D. 50°
**Think:** Since P and F are the midpoints of BD and CD, respectively, we have PF = 0.5BC. Similarly, we can obtain PE = 0.5AD. Since AD = BC, we have PF = PE. Since angle EPF = 130°, we have angle PEF = angle PFE = 0.5 × (180° - 130°) = 25°.
**Answer:** A. 25° |
| | ScienceQA [40] | **Question:** Which better describes the Tibetan Plateau ecosystem?
**Options:** A. It has warm summers. It also has cool winters. B. It has long, cold winters. It also has mostly small plants.
**Think:** A tundra is a type of ecosystem. Tundras have the following features: long, cold winters and short, cold summers, soil that is frozen year-round, and mostly small plants. So, the Tibetan Plateau has long, cold winters. It also has mostly small plants.
**Answer:** B. It has long, cold winters. It also has mostly small plants. |
| Spatial Clue | GOT [24] | **Question:** Please track the motorcycle wheel in the video, its initial coordinates are [107, 54, 159, 82]
**Think:** The video depicts a scene of a motorcycle passing by. The tire of the motorcycle first moves to the upper left and then rotates nearby.
**Clue:** [107, 54, 159, 82], [122, 59, 161, 86], [126, 61, 169, 86], [136, 69, 179, 92] |
| | RefCOCO [72, 41] | **Question:** Detect the giant doughnut with white icing and red , white , and blue sprinkles.
**Think:** The giant doughnut with white icing and red, white, and blue sprinkles is clearly visible on the right side of the tray held by the man. It is distinct from the chocolate-frosted doughnut on the left.
**Clue:** [334.72, 298.08, 522.88, 450.23999999999995] |
| | VisualCoT [47] | **Question:** Can you tell me about the hairstyles of the individuals in the image?
**Options:** A. They have shaggy hair, B. They have perfectly straight hair. C.They have short cropped hair. D. They have very short buzz cuts.
**Think:** The image depicts a casual garden scene, making shaggy hair (Option A) the most fitting choice due to its relaxed, low-maintenance style. Perfectly straight or short, structured cuts (Options B–D) are less suited to such informal outdoor settings.
**Clue:** [98,114,240,146]
**Answer:** A. They have shaggy hair. |

## 7.2 Dataset Statistics

The VTTS-80K dataset is designed to support diverse multimodal reasoning tasks, with varying annotation types across its data sources. In the dataset, all entries include Question and Think annotations, which form the foundational components for reasoning and iterative perception. However, the specific annotation types differ depending on the source of the data.

Data from three sources—GOT [24], Charades [19], and RefCOCO [72, 41]—are primarily of the grounding or tracking type. In these datasets, the task involves directly identifying the relevant clue (either temporal or spatial) within the input, and as such, they only provide clue annotations without accompanying QA pairs. On the other hand, the remaining datasets follow a QA format, where the primary focus is on answering questions based on visual and textual inputs. Among these QA datasets, NextGQA [63] and VisualCoT [47] provide both QA annotations and clue annotations , enabling joint reasoning over questions and supporting evidence. In contrast, the rest of the datasets contain only QA annotations , lacking explicit clue information.

In total, the VTTS-80K dataset comprises 15K temporal clues , 30K spatial clues , 80K Think annotations , and 50K QA pairs . This diverse composition ensures that the dataset supports a wide range of tasks, from fine-grained spatiotemporal localization to complex reasoning over multimodal inputs. By incorporating data with varying levels of annotation richness, VTTS-80K not only

facilitates training models to handle incomplete or heterogeneous data but also reflects real-world scenarios where annotation availability may vary significantly.

# 8 Training Details

The VTTS RL training is configured with the following parameters. We use an AdamW optimizer with a learning rate of $2 \times 10^{-6}$, zero weight decay, and a linear learning rate schedule without warmup. The total batch size is set to 16. For video inputs, the number of frames ranges from 4 to 768 at a fixed frame rate of 2 FPS, with video resolution constrained by a maximum pixel count of $768 \times 28 \times 28$ and a minimum of $128 \times 28 \times 28$. In terms of image settings, the image factor is set to 28, with a maximum scaling ratio of 200. The image resolution follows the same pixel constraints as the video input, ranging from a minimum of $4 \times 28 \times 28$ to a maximum of $768 \times 28 \times 28$.

# 9 Case Study

As shown in Fig. 6 and 7, our Iterative Perception (ITP) approach demonstrates strong performance on both video QA and image reasoning tasks. In the two cases presented, the model initially fails to provide the correct answer during the first round of reasoning. However, through iterative perception, the model successfully identifies the critical clues required to answer the questions. By leveraging the multi-step "think" process, which integrates insights from multiple perception iterations, the model ultimately arrives at the correct answers. This highlights the effectiveness of iterative refinement in enhancing the model's ability to locate relevant information and reason more accurately over complex multimodal inputs.

As shown in Fig 8, 9, 10, 11, and 12, our model demonstrates strong performance across a variety of visual tasks, including grounded video QA, temporal grounding, grounded image QA, spatial grounding, and tracking. In these fine-grained spatiotemporal perception tasks, the model not only accurately provides the perceived temporal and spatial results but also generates well-reasoned explanations for its decisions. This dual capability highlights the model's robust reasoning and spatiotemporal perception abilities, showcasing its proficiency in handling complex, multimodal inputs while maintaining interpretability and precision.

**Failure Cases.** Fig. 13 and 14 presents two failure cases of the model, corresponding to scenarios where the model either consistently provides incorrect answers across multiple perception steps or initially answers incorrectly but later corrects itself. Through analysis of these cases, we observe that the model's failures are primarily due to inaccurate localization of critical clues during the identification process. For knowledge-based questions, the model struggles to pinpoint precise supporting evidence, which can lead to errors in reasoning. Additionally, the model may lose some global contextual information while focusing on localized clues, further contributing to its inability to arrive at the correct answer. These findings underscore the challenges of balancing fine-grained localization with holistic understanding in complex multimodal tasks.

# 10 Discussions

**Limitations.** Despite the effectiveness of our proposed method, several limitations remain that warrant further investigation. First, the iterative nature of our approach may lead to increased inference time, as each additional perception step requires further computation. While this enhances performance, it could pose challenges for real-time applications or scenarios with strict latency constraints. Second, the "thinking" process inherent in iterative perception introduces potential safety concerns. Specifically, intermediate reasoning steps may generate unintended or inappropriate content, which could propagate into the final output. Such issues highlight the need for careful design and safeguards to ensure the reliability and safety of the model in practical deployments.

**Broader Impacts.** Our VTTS method for MLLMs introduces significant advancements in multimodal reasoning but also raises important societal considerations, as outlined under the NeurIPS Code of Ethics.

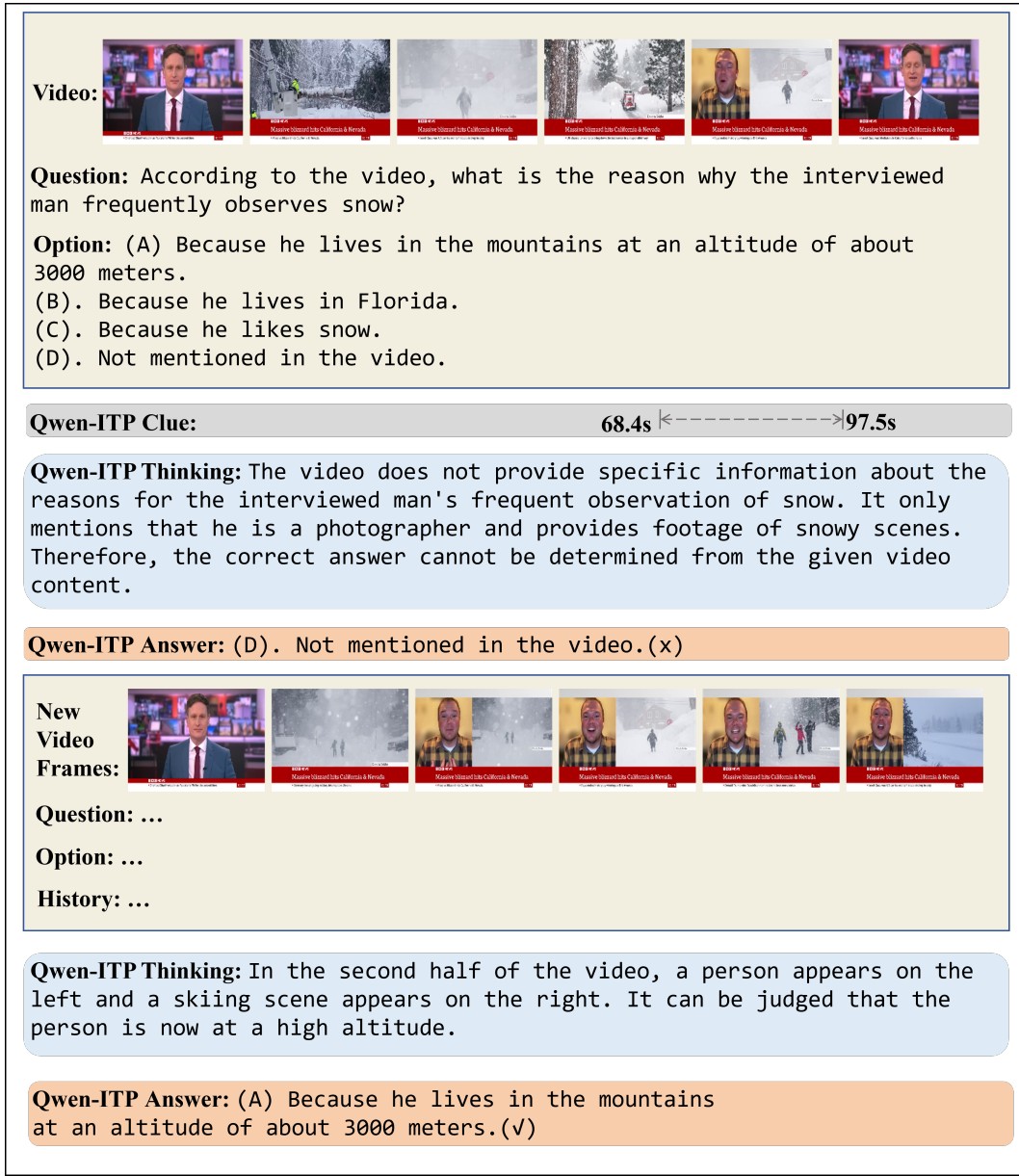

Figure 6: **Iteration Perception For Video.**

VTTS significantly enhances multimodal reasoning by improving iterative visual perception, enabling AI systems to interact with the world more accurately and in a human-like manner. This advancement translates into superior performance across diverse tasks, including video conversation, image reasoning, and spatio-temporal perception, offering benefits in accessibility (e.g., assistive technologies for visually impaired individuals), education (e.g., intelligent tools for complex subjects), safety (e.g., surveillance and autonomous systems), and content creation (e.g., video summarization and moderation). Furthermore, by introducing a novel test-time scaling approach and the VTTS-80K dataset, this work lays a strong foundation for future research, paving the way for the development of adaptive and efficient multimodal AI systems.

VTTS also introduces several potential societal risks, reflecting broader challenges associated with advanced AI systems. A significant concern is the heightened potential for misinformation and deepfakes, as the enhanced ability to generate and manipulate visual and textual content could be exploited to create deceptive or harmful media, such as disinformation campaigns or fabricated

evidence. Additionally, the risk of hallucination—where models generate plausible but incorrect or unsupported outputs—poses further challenges, potentially leading to misleading conclusions in critical applications like news reporting, scientific analysis, or legal contexts. Privacy concerns also arise due to the model's capacity to process detailed visual information, which could be misused in surveillance, unauthorized data collection, or intrusive monitoring scenarios. Moreover, biases inherent in training data may be amplified by VTTS's improved reasoning capabilities, resulting in unfair or discriminatory outcomes in real-world applications, such as biased decision-making in hiring, law enforcement, or healthcare.

**Safeguards.** To address these risks, we adhere to the NeurIPS Code of Ethics, emphasizing responsible research practices and acknowledging the need to consider and mitigate potential harms. Although no specific technical safeguards against deepfake generation are outlined for VTTS, the broader AI community's efforts in detection tools and responsible release strategies provide relevant mitigation pathways. Privacy concerns are addressed through a commitment to ethical guidelines and the integration of privacy-preserving techniques in future deployments. To tackle bias and fairness issues, rigorous dataset auditing and bias mitigation practices are essential.

**Image:**

**Question:** Which of the following is NOT true regarding the divided Germany shown on the map?

**Option:** (A): The capital of West Germany was located in East Germany
(B): West Germany experienced a postwar economic miracle.
(C): Both East Germany and West Germany were allowed to rearm
(D): France had occupied West Germany.

**Qwen-ITP Clue:** [156, 209, 264, 437]

**Qwen-ITP Thinking:** A. The capital of West Germany was located in East Germany - This is not true. Bonn, the capital of West Germany, is clearly shown on the map as being west of the division line.\nB. West Germany experienced a postwar economic miracle - This is true. The economic recovery of West Germany after World War II is well-documented.\nC. Both East Germany and West Germany were allowed to rearm - This is not true. The map shows East Germany as part of the Soviet zone, which would have been under Soviet control and not allowed to rearm independently.\nD. France had occupied West Germany - This is not true. The map does not show any French occupation of West Germany; it is clearly marked as West Germany.

**Qwen-ITP Answer:** (C): Both East Germany and West Germany were allowed to rearm(x)

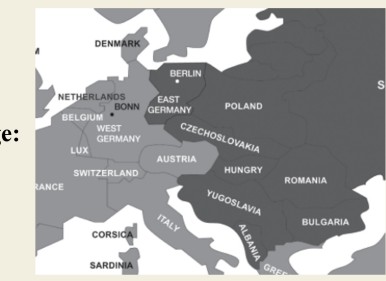

**Image:**

**Question:** …

**Option:** …

**History:** …

**Qwen-ITP Thinking:** (A): The capital of West Germany was located in East Germany. This is false. The capital of West Germany was Bonn (marked on the map within West Germany), while East Germany's capital was East Berlin.(B): West Germany experienced a postwar economic miracle True. West Germany's rapid economic recovery in the 1950s (the Wirtschaftswunder ) is well-documented.(C): Both East Germany and West Germany were allowed to rearm True. West Germany joined NATO and rearmed under U.S. oversight, while East Germany formed the National People's Army under Soviet influence.(D): France had occupied West GermanyTrue. France occupied the southwestern zone of West Germany as part of the Allied post-WWII occupation.

**Qwen-ITP Answer:** (A): The capital of West Germany was located in East Germany(√)

Figure 7: **Iteration Perception For Image.**

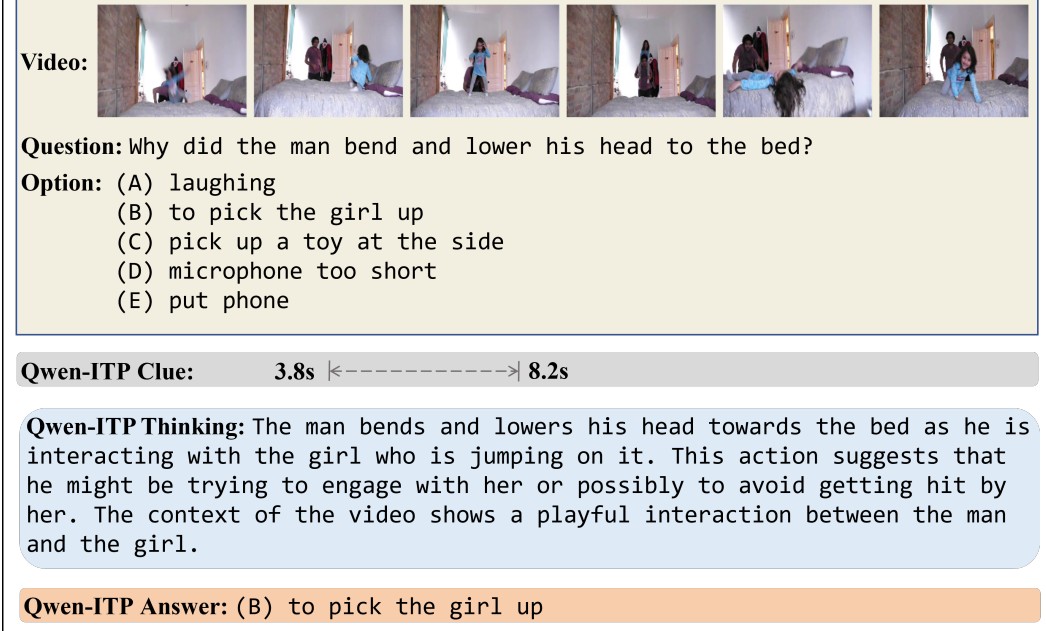

**Video:**

**Question:** Why did the man bend and lower his head to the bed?

**Option:** (A) laughing
(B) to pick the girl up
(C) pick up a toy at the side
(D) microphone too short
(E) put phone

**Qwen-ITP Clue:** 3.8s |←-----------→| 8.2s

**Qwen-ITP Thinking:** The man bends and lowers his head towards the bed as he is interacting with the girl who is jumping on it. This action suggests that he might be trying to engage with her or possibly to avoid getting hit by her. The context of the video shows a playful interaction between the man and the girl.

**Qwen-ITP Answer:** (B) to pick the girl up

Figure 8: **Example For Grounded VideoQA.**

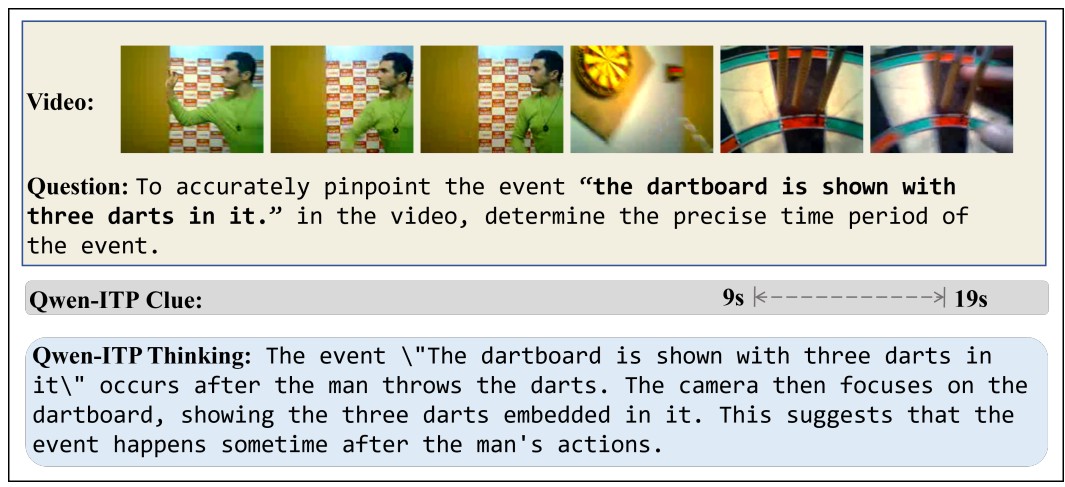

**Video:**

**Question:** To accurately pinpoint the event **"the dartboard is shown with three darts in it."** in the video, determine the precise time period of the event.

**Qwen-ITP Clue:** 9s |←-----------→| 19s

**Qwen-ITP Thinking:** The event \"The dartboard is shown with three darts in it\" occurs after the man throws the darts. The camera then focuses on the dartboard, showing the three darts embedded in it. This suggests that the event happens sometime after the man's actions.

Figure 9: **Example For Temporal Grounding.**

**Image:** 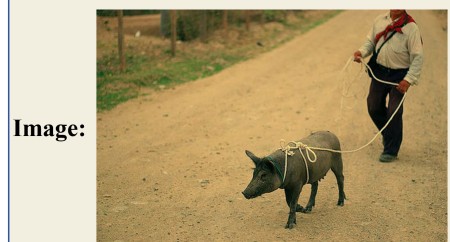

**Question:** What is the person in the image using to guide the animal?",

**Option:** (A):The person is using their hands to guide the animal.
(B):The person is using a wooden stick to guide the animal.
(C):The person is using a long piece of rope to guide the animal.
(D):The person is using a short piece of string to guide the animal.

**Qwen-ITP Clue:** [251, 60, 449, 256]

**Qwen-ITP Thinking:** The person in the image is guiding the animal using a long piece of rope. The pig is tied with a rope around its neck, and the individual is holding the other end of the rope, which appears to be several feet long, allowing them to control the pig's movement without direct physical contact.

**Qwen-ITP Answer:** (C):The person is using a long piece of rope to guide the animal.

Figure 10: **Example For Grounded ImageQA.**

**Image:** 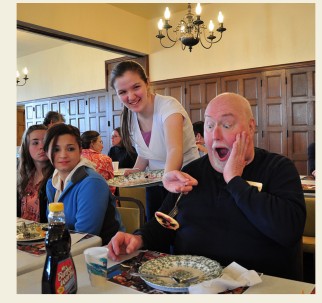

**Question:** Please detect woman most left pink shirt.

**Qwen-ITP Clue:** [0, 259, 70, 480]

**Qwen-ITP Thinking:** The individuals visible from left to right include:A woman with dark hair wearing a blue jacket (over a black shirt).A woman with light brown hair wearing a white shirt. A bald man in a black shirt. A standing woman with blonde hair wearing a white shirt.

Figure 11: **Example For Detection.**

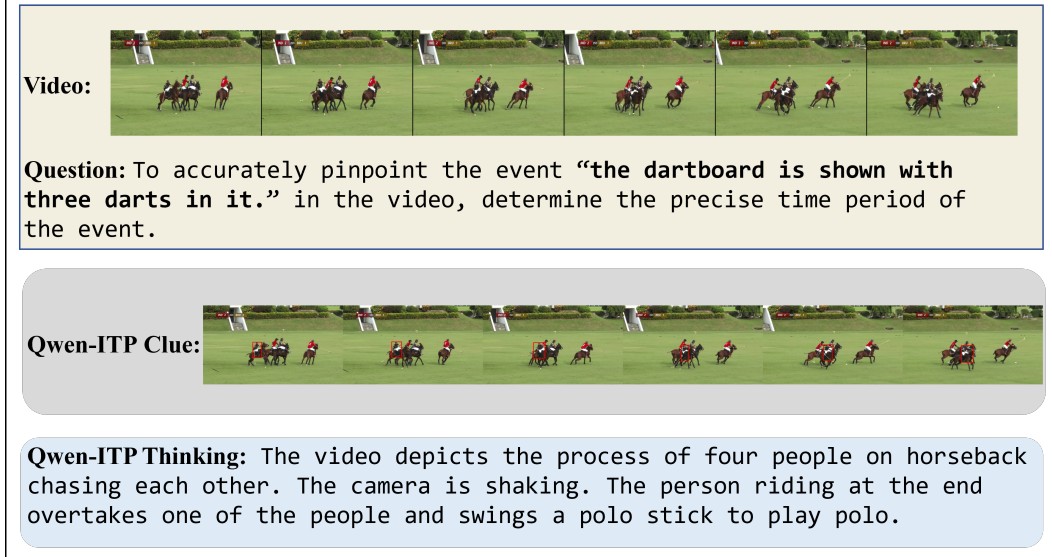

**Video:**

**Question:** To accurately pinpoint the event **"the dartboard is shown with three darts in it."** in the video, determine the precise time period of the event.

**Qwen-ITP Clue:**

**Qwen-ITP Thinking:** The video depicts the process of four people on horseback chasing each other. The camera is shaking. The person riding at the end overtakes one of the people and swings a polo stick to play polo.

Figure 12: **Example For Tracking.**

**Video:** 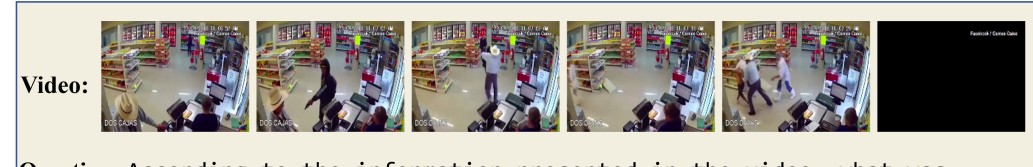

**Question:** According to the information presented in the video, what was the exact number of individuals present in the store at the time?

**Option:** (A). 3
(B). 4
(C). 5
(D). 6

**Qwen-ITP Clue:**    0s ⟵- - - - - - - - - - - - - - - - - - - - - - ⟶    48s

**Qwen-ITP Thinking:** The video shows three individuals: one man wearing a cowboy hat and glasses, another man in a white shirt, and a third person who is not clearly visible but appears to be a customer or employee. The man in the cowboy hat is seen taking off his glasses and pointing a gun, while the other two individuals are seen running away. Therefore, there were three individuals present in the store at the time.

**Qwen-ITP Answer:** (A). 3.(x)

**New Video Frames:**

**Question:** …

**Option:** …

**History:** …

**Qwen-ITP Thinking:** At the beginning of the video, there are two individuals visible: one wearing a cowboy hat and another standing behind the counter. As the video progresses, more people enter the scene, including a man in a white shirt who appears to be involved in an altercation. The total count of individuals in the store at any given moment is not clear from the provided frames alone, but it seems to exceed four due to the presence of multiple people moving around. However, without precise timestamps for each individual's entry and exit, it's challenging to provide an exact number. Given the options, the closest estimate would be B, as it suggests a number greater than four.

**Qwen-ITP Answer:** (B) 4.(x)

Figure 13: **Failure Case For Video Iteration Perception**

**Image:** 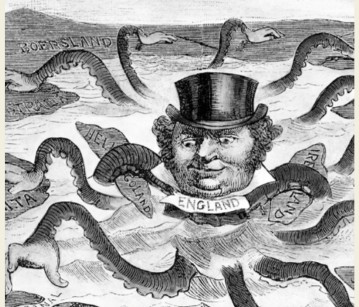

"The Devilfish in Egyptian Waters," 1882

**Question:** A similar cartoon could be made about each of the following countries prior to 1890 EXCEPT?

**Option:** (A): Netherlands.
(B): France.
(C): Belgium.
(D): Germany.

**Qwen-ITP Clue:** [207, 426, 310, 622]

**Qwen-ITP Thinking:** The image depicts a cartoon from 1882 that portrays England as a powerful entity with tentacles extending across various regions, symbolizing its influence and control over these areas. The question asks which country among the given options (Netherlands, France, Belgium, Germany) would not have been depicted similarly due to the historical context before 1890. Given the time frame, it is unlikely that any of these countries would have been portrayed in such a manner because they were not as dominant or influential as England at that time. However, the question specifically asks for an exception, implying that one of these countries might have been depicted differently.

**Qwen-ITP Answer:** (D): Germany(√)

**Image:**

**Question:** …

**Option:** …

**History:** …

**Qwen-ITP Thinking:** The image is a political cartoon depicting England (symbolized by a figure with a top hat and "ENGLAND" on his belt) with tentacles extending to various colonies, illustrating British imperialism. The question asks which country prior to 1890 could not have a similar cartoon made about it, given their colonial activities. While Belgium established the Congo Free State in 1885 (just before 1890), its colonial presence was limited primarily to the Congo. Unlike the other options, Belgium lacked the multiple colonies required to justify a cartoon with numerous "tentacles" akin to the one shown.

**Qwen-ITP Answer:** (C): Belgium(x)

Figure 14: **Failure Case For Image Iteration Perception**

