# OpenReview forum: "VideoChat-R1.5: Visual Test-Time Scaling to Reinforce Multimodal Reasoning by Iterative Perception"
_NeurIPS.cc/2025/Conference — NeurIPS 2025 poster_

### Official Review · Reviewer_DqAN · 2025-07-02

**Clarity:** 3
**Significance:** 3
**Originality:** 3
**Rating:** 4
**Confidence:** 3

**Summary:**

VTTS lets an MLLM re-perceive its visual input several times during inference: after each reasoning step the model predicts a region of interest, crops/resamples that patch, and feeds it back through the vision encoder, with a reinforcement-learned policy (trained via GRPO on the new VTTS-80K dataset) guiding which patches to attend to. Added to Qwen 2.5-VL (3 B and 7 B) this iterative perception loop yields 5–6% average accuracy gains on more than twenty image- and video-reasoning benchmarks, with three iterations giving the best compute-accuracy trade-off.

**Questions:**

- Could the authors elaborate, possibly with a formal discussion or empirical study, why reinforcement learning as implemented here succeeds where supervised fine-tuning does not in narrowing down regions of interest? Is there a formal intuition or guarantee about convergence or optimality?

- What are the practical computation costs (e.g., latency, GPU-hours) associated with 3+ iterations of ITP at inference? Is there a diminishing returns point, and how should practitioners select the right trade-off for deployment?

**Ethical Concerns:**

["NO or VERY MINOR ethics concerns only"]

**Final Justification:**

The authors have addressed most of my concerns with additional experiments, which have improved the overall quality of the paper.

**Limitations:**

yes

**Quality:**

3

**Strengths And Weaknesses:**

### Strengths

- S1: The work tackles the significant challenge of scaling multimodal reasoning in MLLMs via iterative, adaptive computation, grounded in cognitive inspiration (humans' hierarchical attention).

- S2: The core proposal of VTTS with ITP and reinforcement learning is a meaningful extension beyond static perception, modeling perceptual dependencies through reinforcement signals tied to spatio-temporal cues.

- S3: VTTS-80K is a well-motivated resource, with multi-granular annotations (QA, spatio-temporal support, stepwise CoT), systematically constructed as seen in Figure 3, supporting both training and benchmarking of iterative visual reasoning.

- S4: The evaluation covers a wide variety of tasks (video, image, spatial, temporal) and provides quantitative evidence for improvements.

### Weaknesses

- W1:  While the model is compared to strong open-source and some proprietary baselines, some result tables (e.g., Table 7 for spatial grounding) hint that specialized detection and grounding models (e.g., Grounding-DINO-L) are stronger in certain settings. The discussion of tradeoffs versus domain-specific approaches could be more critical—are there settings where VTTS-based models are still weaker?

- W2: The computation and efficiency analysis are missing, such as inference-time FLOPs or latency. Iterative perception may be costly, especially on long videos.

- W3: Limited backbone diversity. Results are only on Qwen-VL 3 B/7 B. Claims of generality would be stronger with models that use different vision encoders.

- W4: No head-to-head with other vision-side TTS baselines such as Visual-CoT or V*, thus the gains cannot be cleanly attributed to the new components. Another straightforward training-free baseline (or ablation study) is sampling $k$ random spatial (or temporal) crops from the input image/video, and fusing the answers by a simple majority vote.

- W5: Inconsistent tense and plurality could be improved for better clarity and professional polish.

---

> ### Author Rebuttal · Authors · 2025-07-31
>
> We sincerely appreciate the reviewer's thoughtful feedback and recognition of our work's significance in scaling multimodal reasoning through iterative, adaptive computation. Thank you for positively evaluating our VTTS approach with ITP and reinforcement learning, as well as acknowledging VTTS-80K as a valuable resource with comprehensive annotations. We are grateful for your appreciation of our diverse task evaluation and quantitative improvements. Your constructive feedback has strengthened our work, and we thank you for your careful consideration. Below, we address each of your comments and questions in detail.
>
>
>
> Table 1: Trade-off Between VTTS Performance and Inference Time
>
> | Model                  | Inference Type | Average A100 Infer Time on MMVet(Image) | Performance on MMVet | Average A100 Infer Time on VideoMME(Video) | Performance on VideoMME |
> | ---------------------- | -------------- | --------------------------------------- | -------------------- | ------------------------------------------ | ----------------------- |
> | Insight-V              | CoT            | 1.32s                                   | 64.7                 | -                                          | -                       |
> | MM-Eureka              | CoT            | 1.39s                                   | 65.9                 | -                                          | -                       |
> | VideoChat-R1           | CoT            | -                                       | -                    | 13.63s                                     | 62.4                    |
> | Qwen2.5VL-7B(Baseline) | Direct Output  | 1.08s                                   | 64.9                 | 11.26s                                     | 64.4                    |
> | Qwen2.5VL-7B(Baseline) | CoT            | 1.46s                                   | 63.2                 | 14.90s                                     | 61.3                    |
> | VTTS-7B                | Single Infer   | 1.35s                                   | 67.2                 | 13.21s                                     | 65.2                    |
> | VTTS-7B                | Multi Infer    | 2.97s                                   | **68.3**             | 30.09s                                     | **67.1**                |
>
> > **Questions 1: Why is the performance of sft worse than that of GRPO?**
>
> While both SFT and GRPO aim to train models to identify implicit clues, GRPO’s use of L1 loss—versus SFT’s cross-entropy loss—proves more effective for continuous or numerical clues (e.g., time stamps, durations, counts). Cross-entropy treats outputs categorically, so predictions like 20.1 and 21.0 incur similar penalties when the ground truth is 20.0, even though 20.1 is semantically much closer. In contrast, L1 loss in GRPO directly penalizes small deviations, enabling fine-grained, metric-aware learning. This leads to more accurate clue extraction, especially in temporal or quantitative reasoning, where precise estimation is critical for downstream perception. As a result, GRPO consistently outperforms SFT, which can degrade performance due to its coarse, discrete supervision on inherently continuous clues.
>
> > **Questions 2: The right trade-off for deployment**
>
> We provide an ablation experiment in Figure 4 in the main text, discussing that as the number of perceptions increases, we find that the second round of multi inference has the greatest benefit, and the benefits thereafter gradually decrease, with almost no additional benefit after the third or fourth rounds.
>
> > **Weakness 1:  Use SOTA expert models for clue finding**
>
> As shown in Table 2, in the QA task, the clue finding method of VTTS is stronger than the traditional model. The reason is our method specifically targets implicit clues that are not directly detectable by standard perception models, which typically recognize explicit entities or actions but fail to reason about contextual or temporal dependencies. In contrast, our approach leverages the rich prior knowledge of VLMs to interpret these implicit clues and guide effective clue extraction, non-training strategies perform poorly in such cases.
>
> Table 2: Abaltion on Clue finding models
>
> | Clue Finding                        | Inference Type | VideoMME |
> | ----------------------------------- | -------------- | -------- |
> | VTTS                                | Single Infer   | 65.2     |
> | Random                              | Multi infer    | 61.0     |
> | SG-DETR(sota in Temporal Grounding) | Multi Infer    | 64.0     |
> | VTTS                                | Multi Infer    | 67.1     |
>
>
>
> >  **Weakness 2: Inference Time of  VTTS.**
>
>  As shown in Table 1, our model has similar inference costs as other CoT models in single infer mode, while its performance is improved and outperforms other CoT models. In multi infer mode, although the inference time is longer, the performance is further improved.
>
> > **Weakness 3: More Results on other MLLMs.**
>
> As shown in Table 1, we have conducted preliminary experiments applying VTTS to InternVL3-8B, using the same VTTS-80K training set to obtain InternVL3-8B-VTTS. The results are summarized in Table 1 below:
>
> Table 3: Results on InternVL3
>
> | Model             | Inference Type | VideoMME |
> | ----------------- | -------------- | -------- |
> | InternVL3-8B      | Direct Output  | 65.3     |
> | InternVL3-8B-VTTS | Single Infer   | 65.9     |
> | InternVL3-8B-VTTS | Multi Infer    | 66.7     |
>
> These results show that VTTS consistently improves performance on InternVL3-8B under both single and multiple inference settings. This indicates that our method is not limited to Qwen2.5-VL and can generalize to other strong MLLMs with different architectures and training paradigms.
>
> At the same time, we found during the training process that InternVL3 is worse than Qwen in terms of instruction following ability, and the preference disappears during RL training, so the performance improvement of InternVL3 is less than that of Qwen.
>
>
>
> > **Weakness 4: More Results on Vision-side TTS baselines**
>
> In Table 1, we provide more experimental results. You can see that our model has improved capabilities compared to visual reasoning models such as Insight-V and MM-Eureka. Since the base model used by V* and Visual COT is relatively weak, we did not compare them. At the same time, in Table 2, we provide a baseline for random clue selection. You can see that VTTs is more capable of selecting clues than random and expert models.
>
> > **Weakness 5: Problems in writing**
>
> We thank the reviewer for the feedback. In the camera-ready version, we will carefully proofread the paper to correct inconsistent tense and plurality, ensuring improved clarity and professional presentation.

---

> > ### Comment · Reviewer_DqAN · 2025-08-06
> >
> > I appreciate the authors' efforts in addressing most of my concerns and the additional supportive results. I would keep the positive score.

---

### Official Review · Reviewer_DWnj · 2025-07-03

**Clarity:** 3
**Significance:** 3
**Originality:** 3
**Rating:** 5
**Confidence:** 4

**Summary:**

This paper introduces VTTS (Visual Test-Time Scaling), a method to enhance reasoning in Multimodal Large Language Models (MLLMs) via iterative visual perception at inference time. Inspired by human-like progressive attention, VTTS refines attention to high-confidence spatio-temporal regions in multiple rounds, driven by evolving textual predictions. It proposes: 1) An Iterative Perception (ITP) mechanism using Reinforcement Learning (RL) based on GRPO. 2) A new dataset, VTTS-80K, designed for training MLLMs with spatio-temporal grounding and reasoning via RL. 3) Extensive experiments showing 5–6% performance boosts over strong MLLM baselines across more than 20 benchmarks, including video QA, image reasoning, and perception.

**Questions:**

* Heavy Reliance on Qwen2.5-VL: (no worries if this may take some time and cannot be completed during rebuttal, I can understand it.) All model improvements are tied to Qwen2.5; generalization to other MLLMs (e.g., LLaVA, Gemini) may need to be demonstrated. Could authors have a try on other frameworks if possible?

I appreciate the completeness of this system paper, which demonstrates a well-integrated effort across dataset construction, model architecture design, and comprehensive empirical evaluation. However, I have concerns regarding the generalizability of the proposed algorithm and would appreciate further clarification on the disentanglement between visual scaling and language-level scaling effects.

**Ethical Concerns:**

["NO or VERY MINOR ethics concerns only"]

**Final Justification:**

I appreciate the authors for addressing most of concerns during rebuttal. Therefore, i will keep my initial score and recommend the acceptance of the paper.

**Limitations:**

Yes.

**Paper Formatting Concerns:**

No.

**Quality:**

3

**Strengths And Weaknesses:**

### **Strengths**

* VTTS introduces a principled way to scale perceptual computation at test time, rather than only scaling generation or language reasoning.
* Demonstrates consistent improvements (1–6% gains) across a wide range of tasks and benchmarks using Qwen2.5-VL models.
* New Benchmark Dataset: VTTS-80K provides high-quality, multi-level annotations (QA, spatio-temporal regions, and CoT) for robust training.
* Detailed Ablation Studies: Includes insightful ablations showing impact of perception steps, data components, and training methods.

---

### **Weaknesses**

* Disentangling Visual Scaling from Language Scaling: Given that language reasoning can also benefit from more decoding steps (e.g., Best-of-N, beam search), how do the authors ensure that the performance gains from VTTS are due to visual test-time scaling rather than indirectly benefiting from repeated language generation?
* The paper models the ROI selection process as a reinforcement learning problem using GRPO. Could the authors elaborate on the nature of the learned spatio-temporal policy? Is the policy generalizable across tasks, or is it overfit to the specific structure of VTTS-80K?
* Engineering cost at inference is high: Iterative perception increases test-time compute, which might be impractical for real-time applications without optimization.
* Quality of VTTS-80K may be bottlenecked by the capabilities of DeepSeek or other LLMs used in the annotation process. It is a concern that dataset generation pipeline heavily relies on LLMs.

---

> ### Author Rebuttal · Authors · 2025-07-30
>
> We sincerely thank the reviewers for their valuable feedback and insightful comments. We appreciate the reviewers' recognition of our novel VTTS approach for scaling perceptual computation at test time, the consistent performance improvements across diverse tasks using Qwen2.5-VL models, our high-quality VTTS-80K benchmark dataset with comprehensive multi-level annotations, and our thorough ablation studies providing meaningful insights into various components. Below, we address each of your comments and questions in detail.
>
> > **Questions 1: More Results on Other MLLM**
>
> As suggested, we have conducted preliminary experiments applying VTTS to InternVL3-8B, using the same VTTS-80K training set to obtain InternVL3-8B-VTTS. The results are summarized in Table 1 below:
>
> Table 1: Results on InternVL3
>
> | Model             | Inference Type | VideoMME |
> | ----------------- | -------------- | -------- |
> | InternVL3-8B      | Direct Output  | 65.3     |
> | InternVL3-8B-VTTS | Single Infer   | 65.9     |
> | InternVL3-8B-VTTS | Multi Infer    | 66.7     |
>
> These results show that VTTS consistently improves performance on InternVL3-8B under both single and multiple inference settings. This indicates that our method is not limited to Qwen2.5-VL and can generalize to other strong MLLMs with different architectures and training paradigms.
>
> At the same time, we found during the training process that InternVL3 is worse than Qwen in terms of instruction following ability, and the preference disappears during RL training, so the performance improvement of InternVL3 is less than that of Qwen.
>
>
>
> > **Weakness 1: Gain From Visual Scaling**
>
> Table 2: Abalation Study on Visual Scaling
>
> | Model     | Inference Type          | VideoMME |
> | --------- | ----------------------- | -------- |
> | Qwen2.5VL | Direct Output           | 64.4     |
> | VTTS      | Single Infer            | 65.2     |
> | VTTS      | Multi infer + only text | 65.9     |
> | VTTS      | Multi Infer + visual    | **67.1** |
>
> To disentangle the effect of visual scaling from repeated language generation, we ablate VTTS under different inference settings in Table 2. While multi-inference with text-only aggregation yields a modest gain (+0.7% over single infer), the full VTTS approach with visual resampling achieves a +1.9% improvement, outperforming text-only multi-inference by +1.2%. This clear performance gap demonstrates that the gains primarily stem from visual test-time scaling—i.e., leveraging diverse visual representations—rather than repeated language generation alone.
>
>
>
> > **Weakness 2: The essence of the ROI selection process**
>
> Our ROI policy is designed to capture spatio-temporal implicit cues in response to questions, leveraging the rich priors of VLMs to interpret and localize such cues. While trained on VTTS-80K, the policy learns task-agnostic reasoning patterns—i.e., how to find relevant spatio-temporal evidence given a question—enabling generalization across diverse downstream tasks rather than overfitting to dataset-specific structures.
>
> > **Weakness 3: High Inference Cost**
>
> We acknowledge that iterative perception increases inference costs. In future work, we plan to optimize engineering efficiency by caching previously computed visual tags and other inference optimizations to significantly reduce redundant computations and improve runtime performance.
>
> > **Weakness 4: Quality of VTTS-80K**
>
> We acknowledge the potential limitations imposed by LLMs in the annotation pipeline. However, we employ careful filtering, multi-round verification to ensure high-quality annotations. Moreover, the spatiotemporal cues and QA of the dataset are partially derived from the original annotations, so the search for implicit spatiotemporal capabilities is not limited to LLMs.

---

> > ### Comment · Reviewer_DWnj · 2025-08-06
> >
> > I appreciate the authors for addressing most of concerns during rebuttal. I think the authors have addressed most of my concerns in the rebuttal.
> >
> > The authors acknowledge that iterative perception increases inference cost. To better understand the practical implications, could the authors provide latency comparisons on the evaluation datasets?

---

> ### Author Response · Authors · 2025-08-07
> **Response to reviewer DWnj07 : Latency Comparisons on the Evaluation Datasets**
>
> Thank you for your reply. We provide the trade-off between inference latency and performance in the table below.
>
> Table 1: Trade-off Between VTTS Performance and Inference Time
>
> | Model                  | Inference Type | Average A100 Infer Time on MMVet(Image) | Performance on MMVet | Average A100 Infer Time on VideoMME(Video) | Performance on VideoMME |
> | ---------------------- | -------------- | --------------------------------------- | -------------------- | ------------------------------------------ | ----------------------- |
> | Insight-V              | CoT            | 1.32s                                   | 64.7                 | -                                          | -                       |
> | MM-Eureka              | CoT            | 1.39s                                   | 65.9                 | -                                          | -                       |
> | VideoChat-R1           | CoT            | -                                       | -                    | 13.63s                                     | 62.4                    |
> | Qwen2.5VL-7B(Baseline) | Direct Output  | 1.08s                                   | 64.9                 | 11.26s                                     | 64.4                    |
> | Qwen2.5VL-7B(Baseline) | CoT            | 1.46s                                   | 63.2                 | 14.90s                                     | 61.3                    |
> | VTTS-7B                | Single Infer   | 1.35s                                   | 67.2                 | 13.21s                                     | 65.2                    |
> | VTTS-7B                | Multi Infer    | 2.97s                                   | **68.3**             | 30.09s                                     | **67.1**                |
>
>
>
> Our model incurs inference costs comparable to other TTS models in single inference mode, while achieving superior performance. In multi-inference mode, although the computational time increases, we observe further performance gains, demonstrating the effectiveness of our approach in leveraging additional reasoning steps. This trade-off between inference cost and accuracy provides a favorable balance, particularly in settings where performance is prioritized.

---

### Official Review · Reviewer_DAuW · 2025-07-03

**Clarity:** 3
**Significance:** 2
**Originality:** 2
**Rating:** 4
**Confidence:** 4

**Summary:**

This paper proposes Visual Test-Time Scaling (VTTS), a method that enhances MLLM reasoning by enabling iterative visual perception during inference. Mimicking human attention, VTTS progressively refines focus on key spatio-temporal regions guided by evolving text predictions. It introduces an Iterative Perception (ITP) mechanism, trained via Generalized Reward Policy Optimization (GRPO) with spatio-temporal supervision. This paper also presents VTTS-80K, a dataset tailored for this paradigm. VTTS achieves consistent improvements across 20+ benchmarks.

**Questions:**

1. What would the performance look like if we directly use ground-truth clues for video or image QA? Alternatively, how does it compare to first extracting clues using a state-of-the-art model, followed by perception? How do these non-training strategies differ from trained approaches in terms of both accuracy and computational efficiency?
2. Analyze why Supervised Fine-Tuning (SFT) may degrade performance in some cases, while GRPO can lead to improvements.
3. The method performs poorly when using only the first perception step and requires multiple iterations to achieve good performance. Why is this the case?
4. It takes more time to perceive temporal and spatial clues multiple times. Did the author count the time taken in different situations? How much difference is there in time efficiency between it and the SFT model (directly outputs the answer) and the inference model (single-inference text output)?
5. Did the model undergo the Cold Start before reinforcement training? Did the author do some detailed ablation experiments to show whether there was any impact of cold start?
6. The authors do not provide visualization of the reasoning results of specific instances of the proposed model, including output reasoning paths and visualizations, which makes it difficult to understand how the model performs video or image reasoning.

**Ethical Concerns:**

["NO or VERY MINOR ethics concerns only"]

**Final Justification:**

The authors have addressed most of my concerns by supplementing additional experiments, which have improved the overall quality of the paper. However, several notable limitations remain, preventing a stronger recommendation for acceptance. First, while the method achieves a modest performance improvement, its inference time is three times longer than the baseline, raising concerns about its practical efficiency. Second, the use of a state-of-the-art (SOTA) temporal grounding model unexpectedly leads to degraded performance, and the paper lacks a thorough analysis to explain this counterintuitive result. Additionally, the discussions on supervised fine-tuning (SFT) and GRPO remain somewhat superficial and could benefit from deeper analysis. Given these unresolved issues, I recommend a borderline accept score.

**Limitations:**

yes

**Quality:**

3

**Strengths And Weaknesses:**

Strengths:
1. VTTS mimics human hierarchical attention by iteratively refining focus on high-confidence spatio-temporal regions. This idea makes sense in many scenarios.
2. Extensive and detailed experiments validate the effectiveness of the proposed method across a wide range of tasks, including Multimodal Video Understanding, Multimodal Image Understanding, Grounded Video Question Answering, Temporal Grounding, Spatial Grounding, and Object Tracking.

Weaknesses:

Major Weakness
1. The paper lacks a direct comparison with other state-of-the-art methods built upon the Qwen2.5-VL or other models.
2. In terms of reinforcement learning, only the reward function was modified, with no significant novelty.
3. Compared to the increased inference time of this method, the performance improvement is not significant enough.
4. Although the idea that focuses on the temporal clue and spatial clue makes sense, the performance improvements of Table 1 (video benchmarks) and Table 2 (image benchmarks) are not satisfactory. Why is this the case? Is there any detailed analysis?

Minor weakness
1. Mistakenly wrote "Fine" as "Fune" in table 5.
2.  Mistakenly wrote "Clue" as "Glue" in figure 3.
3. The text in the figures is a bit blurry.

---

> ### Author Rebuttal · Authors · 2025-07-30
>
> Thank you for your valuable feedback and constructive suggestions. We appreciate your recognition that our VTTS framework's core idea of mimicking human hierarchical attention through iterative refinement on high-confidence spatio-temporal regions is conceptually sound and applicable to various scenarios, as well as your acknowledgment of our extensive experimental validation across multiple tasks including Multimodal Video Understanding, Multimodal Image Understanding, Grounded Video Question Answering, Temporal Grounding, Spatial Grounding, and Object Tracking. Below, we address each of your comments and questions in detail.
>
> Table 1: Trade-off Between VTTS Performance and Inference Time
>
> | Model                  | Inference Type | Average A100 Infer Time on MMVet(Image) | Performance on MMVet | Average A100 Infer Time on VideoMME(Video) | Performance on VideoMME |
> | ---------------------- | -------------- | --------------------------------------- | -------------------- | ------------------------------------------ | ----------------------- |
> | Mulberry               | CoT            | 1.26s                                   | 43.9                 | -                                          | -                       |
> | LLaVA-CoT              | CoT            | 1.41s                                   | 60.3                 | -                                          | -                       |
> | Insight-V              | CoT            | 1.32s                                   | 64.7                 | -                                          | -                       |
> | MM-Eureka              | CoT            | 1.39s                                   | 65.9                 | -                                          | -                       |
> | VideoChat-R1           | CoT            | -                                       | -                    | 13.63s                                     | 62.4                    |
> | Qwen2.5VL-7B(Baseline) | Direct Output  | 1.08s                                   | 64.9                 | 11.26s                                     | 64.4                    |
> | Qwen2.5VL-7B(Baseline) | CoT            | 1.46s                                   | 63.2                 | 14.90s                                     | 61.3                    |
> | VTTS-7B                | Single Infer   | 1.35s                                   | 67.2                 | 13.21s                                     | 65.2                    |
> | VTTS-7B                | Multi Infer    | 2.97s                                   | **68.3**             | 30.09s                                     | **67.1**                |
>
> > **Questions 1:  Use SOTA expert models for clue finding**
>
> As shown in Table 2, in the QA task, the clue finding method of VTTS is stronger than the traditional model. The reason is our method specifically targets implicit clues that are not directly detectable by standard perception models, which typically recognize explicit entities or actions but fail to reason about contextual or temporal dependencies. In contrast, our approach leverages the rich prior knowledge of VLMs to interpret these implicit clues and guide effective clue extraction, non-training strategies perform poorly in such cases.
>
> Table 2: Abaltion on Clue finding models
>
> | Clue Finding                        | Inference Type | VideoMME |
> | ----------------------------------- | -------------- | -------- |
> | VTTS                                | Single Infer   | 65.2     |
> | SG-DETR(sota in Temporal Grounding) | Multi Infer    | 64.0     |
> | VTTS                                | Multi Infer    | 67.1     |
>
>
>
> > **Questions 2: Why is the performance of sft worse than that of GRPO?**
>
> While both SFT and GRPO aim to train models to identify implicit clues, GRPO’s use of L1 loss—versus SFT’s cross-entropy loss—proves more effective for continuous or numerical clues (e.g., time stamps, durations, counts). Cross-entropy treats outputs categorically, so predictions like 20.1 and 21.0 incur similar penalties when the ground truth is 20.0, even though 20.1 is semantically much closer. In contrast, L1 loss in GRPO directly penalizes small deviations, enabling fine-grained, metric-aware learning. This leads to more accurate clue extraction, especially in temporal or quantitative reasoning, where precise estimation is critical for downstream perception. As a result, GRPO consistently outperforms SFT, which can degrade performance due to its coarse, discrete supervision on inherently continuous clues.
>
> > **Questions 3: Why is single-step perception not as good as multi-step perception?**
>
> During the initial step, the model samples visual information uniformly without leveraging any learned or predicted visual clues. At this stage, the model lacks sufficient guidance to focus on relevant regions, resulting in suboptimal perception. However, in subsequent iterations, the model can utilize implicit visual clues derived from its own predictions to perform non-uniform, adaptive sampling—effectively refining its focus on more informative regions. This iterative refinement allows the model to progressively improve its perception and prediction accuracy. Therefore, multiple perception steps are essential to fully exploit the learned visual patterns and achieve strong performance.
>
> > **Questions 4: Single and multiple inference time**
>
> We give the inference time of VTTS model and baseline models in Table 1.
>
> > **Questions 5: Cold Start**
>
> In our work, we did not employ a cold start phase before reinforcement learning, as the base Qwen model already exhibits strong instruction-following capabilities and a solid foundation in reasoning. Thanks to this strong pre-training, the model is able to generate partially correct output formats and meaningful reasoning trajectories solely through well-designed prompts, even before fine-tuning. After just a few steps of reinforcement training, the model quickly stabilizes and produces nearly perfect formatting and coherent reasoning chains. This rapid convergence suggests that a separate supervised fine-tuning (SFT) cold start phase is not necessary in our setup, and the policy initialization is sufficiently strong to enable effective reinforcement learning.
>
> > **Questions 6: Visualization**
>
> We have included detailed case studies in the supplementary material, which contain visualizations of the model’s output reasoning paths. These examples illustrate how our model progressively perceives visual content, refines its reasoning, and arrives at the final prediction.
>
> > **Major Weakness 1: More comparisons with other models**
>
> In Table 1, we provide more experimental results, showing that the VTTS model is ahead of other models, including the baseline (videochat-r1) based on qwen2.5vl.
>
> > **Major Weakness 2: Innovations in reinforcement learning**
>
> While our approach does involve modifying the reward function, it enables a novel training paradigm for multi-step perception and reasoning, which goes beyond a simple reward redesign. Specifically, we introduce a multi-reward structure—comprising answer reward, spatiotemporal reward, and format reward. This composite reward signal guides the model to learn not only what the correct output is, but also how to arrive at it through structured, step-by-step reasoning. The integration of these rewards within the reinforcement learning framework facilitates the emergence of interpretable and controllable reasoning behavior, which is not typically achieved through standard reward setups. Thus, our work contributes a meaningful and effective advancement in the application of RL to complex reasoning tasks.
>
> >  **Major Weakness 3: Relationship between performance and inference time**
>
> As shown in Table 1, VTTS strikes a strong balance between efficiency and performance. On MMVet, VTTS-7B (Single Infer) achieves 1.35s inference time—comparable to LLaVA-CoT (1.41s) and MM-Eureka (1.39s), slightly above the baseline (1.08s)—but with significantly higher performance (67.2%). On VideoMME, VTTS-7B (Single Infer) takes 13.21s (vs. 11.26s baseline) yet achieves 65.2%, outperforming existing methods. Thus, VTTS improves performance with acceptable latency, offering a more effective trade-off than current long-chain inference approaches.
>
> > **Major Weakness 4: Ways to continue improving performance**
>
> We show the failure case in Appendix Fig. 9. Despite the design of a reasonable reward function, the model still finds it difficult to accurately locate some spatiotemporal clues. This shows that the current training data and SFT stage are still insufficient in modeling spatiotemporal perception capabilities, and further data expansion and training strategy optimization are needed.
>
> >  **Minor weakness:**
>
> We thank the reviewer for the careful feedback. We acknowledge the minor errors and will correct "Fune" to "Fine" in Table 5 and "Glue" to "Clue" in Figure 3 in the final version. Additionally, we will improve the figure resolution to ensure all text is clear and legible. These revisions will be made promptly in the camera-ready copy.

---

> > ### Comment · Reviewer_DAuW · 2025-08-07
> >
> > The authors' response addresses most of my concerns, so I prefer to accept this paper.

---

### Official Review · Reviewer_T9BA · 2025-07-03

**Clarity:** 3
**Significance:** 2
**Originality:** 2
**Rating:** 4
**Confidence:** 4

**Summary:**

This paper introduces Visual Test-Time Scaling (VTTS), a framework that enhances MLLMs through iterative perception during inference. The core innovation lies in mimicking human hierarchical attention by progressively refining focus on high-confidence spatio-temporal regions, guided by evolving textual predictions. The authors also contribute VTTS-80K, a dataset tailored for iterative perception training. Experiments on video and image understanding show significant gains on Qwen-VL baselines.

**Questions:**

1. What is the computational overhead of the iterative inference, compared with the direct answer baseline and other long-chain inference methods (e.g. LLaVA-CoT, Mulberry, etc.)?
2. The proposed concept of test-time scaling is a type of reasoning method. What is the performance compared with other reasoning post-training (e.g. LLaVA-CoT, Mulberry, Insight-V, MM-Eureka, Visual-CoT, etc) and test-time scaling approaches?

**Ethical Concerns:**

["NO or VERY MINOR ethics concerns only"]

**Final Justification:**

The authors have addressed most of my concerns. Therefore, I decided to raise my point to accept. Authors are encouraged to add the efficiency-performance test to the camera-ready version.

**Limitations:**

Yes

**Quality:**

2

**Strengths And Weaknesses:**

Strengths:
- The proposed concept of iterative perception is reasonable and effective.
- The integration of reinforcement learning (GRPO) with spatio-temporal supervision is well-designed.
- The proposed VTTS datasets are valuable for the research community.

Weaknesses:
- Lack of some essential comparisons and experiments. Please refer to the question period.
- The authors include no sota reasoning approaches in the table for comparison.

---

> ### Author Rebuttal · Authors · 2025-07-30
>
> We sincerely appreciate the reviewer's thoughtful and constructive feedback on our work. We are grateful for the positive recognition of our key contributions, including the reasonableness and effectiveness of our iterative perception method, the well-designed integration of reinforcement learning with spatio-temporal supervision, and the value of our proposed VTTS datasets for the research community. These encouraging comments motivate us to further improve our work and address the remaining concerns raised by the reviewer. Below, we address each of your comments and questions in detail.
>
> Table 1: Trade-off Between VTTS Performance and Inference Time
>
> | Model                  | Inference Type | Average A100 Infer Time on MMVet(Image) | Performance on MMVet | Average A100 Infer Time on VideoMME(Video) | Performance on VideoMME |
> | ---------------------- | -------------- | --------------------------------------- | -------------------- | ------------------------------------------ | ----------------------- |
> | Mulberry               | CoT            | 1.26s                                   | 43.9                 | -                                          | -                       |
> | LLaVA-CoT              | CoT            | 1.41s                                   | 60.3                 | -                                          | -                       |
> | Insight-V              | CoT            | 1.32s                                   | 64.7                 | -                                          | -                       |
> | MM-Eureka              | CoT            | 1.39s                                   | 65.9                 | -                                          | -                       |
> | VideoChat-R1           | CoT            | -                                       | -                    | 13.63s                                     | 62.4                    |
> | Qwen2.5VL-7B(Baseline) | Direct Output  | 1.08s                                   | 64.9                 | 11.26s                                     | 64.4                    |
> | Qwen2.5VL-7B(Baseline) | CoT            | 1.46s                                   | 63.2                 | 14.90s                                     | 61.3                    |
> | VTTS-7B                | Single Infer   | 1.35s                                   | 67.2                 | 13.21s                                     | 65.2                    |
> | VTTS-7B                | Multi Infer    | 2.97s                                   | **68.3**             | 30.09s                                     | **67.1**                |
>
>
>
> > **Reply to Q1: Computational overhead of the iterative inference：**
>
> As shown in Table 1, VTTS achieves a good balance between computational overhead and performance compared to the direct output baseline and other long-chain inference methods. Specifically, VTTS-7B (Single Infer) achieves an average inference time of 1.35 seconds on MMVet, slightly higher than the direct output baseline's 1.08 seconds. This is similar to other long-chain inference methods, such as LLaVA-CoT (1.41 seconds) and MM-Eureka (1.39 seconds), and its performance (67.2%) far exceeds these methods. Furthermore, while multi-inferring increases inference time, it also improves performance by 1.1%. For the video task (VideoMME), VTTS-7B (Single Infer) achieves an inference time of 13.21 seconds, higher than the baseline's 11.26 seconds, yet achieves a performance of 65.2%, significantly outperforming other methods. Therefore, VTTS's Single Infer method improves performance while maintaining similar inference time to existing CoT methods. Meanwhile, Multi-infer achieves significant performance improvements while incurring moderate computational overhead, outperforming most existing long-chain inference methods.
>
>
>
> > **Reply to Q2, W1, W2: Performance compared with other reasoning models:**
>
> As shown in Table 1, VTTS-7B achieved a performance of 68.3% on MMVet (images) and 67.1% on VideoMME (videos) in multi infer mode, significantly outperforming other post-inference training methods (such as LLaVA-CoT, Insight-V, and MM-Eureka) and test-time expansion methods. For example, LLaVA-CoT achieved a performance of 60.3% on MMVet, while Insight-V and MM-Eureka achieved performances of 64.7 and 65.9, respectively, both lower than VTTS. Furthermore, VTTS also outperformed the VideoChat-R1 model (62.4) on video tasks. This performance gain demonstrates the effectiveness of the test-time expansion strategy, indicating that VTTS has stronger reasoning capabilities for complex visual tasks.

---

> > ### Comment · Reviewer_T9BA · 2025-08-05
> >
> > Thanks for the reply, which has addressed most of my concerns.

---

### Note · Authors · 2025-08-12

We thank the reviewers and area chair for their constructive comments and efforts during the review and rebuttal stages.

During the initial review phase, our paper received scores of 5 (DWnj), 4 (DqAN), 4 (DAuW), and 3 (T9BA). The reviewers highlighted several strengths of our work, including:

> Reviewer T9BA: "The proposed concept of iterative perception is reasonable and effective."

> Reviewer DAuW: "VTTS mimics human hierarchical attention by iteratively refining focus on high-confidence spatio-temporal regions. This idea makes sense in many scenarios."

> Reviewer DWnj: "VTTS introduces a principled way to scale perceptual computation at test time, rather than only scaling generation or language reasoning."

> Reviewer DqAN: "The core proposal of VTTS with ITP and reinforcement learning is a meaningful extension beyond static perception, modeling perceptual dependencies through reinforcement signals tied to spatio-temporal cues."

In addition, the reviewers provided numerous constructive suggestions and concerns. During the rebuttal phase, we carefully addressed each point in detail. All concerns raised by the reviewers have been thoroughly responded to, and we are grateful for their positive and encouraging feedback:

> Reviewer T9BA: "Thanks for the reply, which has addressed most of my concerns."

> Reviewer DAuW: "The authors' response addresses most of my concerns, so I prefer to accept this paper."

> Reviewer DWnj: "I appreciate the authors for addressing most of my concerns during rebuttal. I think the authors have addressed most of my concerns in the rebuttal."

> Reviewer DqAN: "I appreciate the authors' efforts in addressing most of my concerns and the additional supportive results. I would keep the positive score."

We will fully incorporate all reviewer suggestions into the final version of the paper, ensuring that revisions are comprehensive, accurate, and clearly reflected in both the main text and the appendix. We are confident that the final manuscript will meet the reviewers’ expectations and further strengthen the contribution of our work.

We sincerely thank the reviewers and the area chair for their time, diligence, and constructive engagement throughout the review process. Their insightful feedback has been instrumental in improving the quality and clarity of our paper.

---

### Decision · Program_Chairs · 2025-09-17

**Decision:**

Accept (poster)

**Comment:**

This paper proposes an inference-time scaling approach to improve visual perception in MLLMs by progressively refining attention on high-confidence spatio-temporal regions, guided by evolving textual predictions. The work also contributes the VTTS-80K dataset, curated for iterative perception training. Experiments on image and video understanding tasks demonstrate improvements over the Qwen-VL baselines.

Some concerns remain, as highlighted by reviewer `DAuW`. These include the heavy compute cost (a 3x increase in inference time over the baseline), degraded performance when integrated with a state-of-the-art temporal grounding model, and the lack of discussion regarding SFT vs. GRPO.

Despite these limitations, all reviewers support accepting the paper (3 borderline accept, 1 accept). After carefully reading the reviews, the rebuttal, and all discussions, I concur with the reviewers' recommendation and suggest this paper be accepted.